# Fatty acid metabolism in aggressive B-cell lymphoma is inhibited by tetraspanin CD37

Rens Peeters [1], Jorge Cuenca-Escalona[1], Esther A. Zaal [2], Anna T. Hoekstra[3], Anouk C. G. Balvert[1], Marcos Vidal-Manrique [1], Niek Blomberg[4], Sjoerd J. van Deventer[1], Rinke Stienstra[5], Julia Jellusova [6,7], Martin Giera [4], Luciana Hannibal[8], Ute Spiekerkoetter [9], Martin ter Beest[1], Celia R. Berkers [2,3] & Annemiek B. van Spriel [1] ✉

The importance of fatty acid (FA) metabolism in cancer is well-established, yet the mechanisms underlying metabolic reprogramming remain elusive. Here, we identify tetraspanin CD37, a prognostic marker for aggressive B-cell lymphoma, as essential membrane-localized inhibitor of FA metabolism. Deletion of CD37 on lymphoma cells results in increased FA oxidation shown by functional assays and metabolomics. Furthermore, CD37-negative lymphomas selectively deplete palmitate from serum in mouse studies. Mechanistically, CD37 inhibits the FA transporter FATP1 through molecular interaction. Consequently, deletion of CD37 induces uptake and processing of exogenous palmitate into energy and essential building blocks for proliferation, and inhibition of FATP1 reverses this phenotype. Large lipid deposits and intracellular lipid droplets are observed in CD37-negative lymphoma tissues of patients. Moreover, inhibition of carnitine palmitoyl transferase 1 A significantly compromises viability and proliferation of CD37-deficient lymphomas. Collectively, our results identify CD37 as a direct gatekeeper of the FA metabolic switch in aggressive B-cell lymphoma.

Aggressive cancers rewire their metabolism to overcome environmental limitations and facilitate rapid proliferation. Already in 1923, Otto Warburg and colleagues proposed that malignant cells undergo a metabolic switch from oxidative respiration to aerobic glycolysis[1]. Regarded as the "Warburg effect", this phenomenon has been observed throughout a broad spectrum of malignancies[2]. However, multiple studies have shown that metabolic reprogramming in cancer cells is not limited to an aerobic glycolytic shift[3]. In fact, fatty acid (FA)

metabolism is the most altered metabolic pathway in a multitude of cancer types and often correlates with poor clinical outcome[4-10]. Transporter-facilitated uptake or synthesis of FAs via de novo lipogenesis is essential to maintain the structural stability of the cellular membranes[11,12]. Furthermore, FAs are important secondary messengers and an excellent source of energy through fatty acid oxidation (FAO or β-oxidation), especially during decreased energy intake or environmental substrate limitations[13]. As such, alterations in FA

[1]Department of Tumor Immunology, Radboud Institute for Molecular Life Sciences, Radboud University Medical Centre, Nijmegen, The Netherlands. [2]Division of Cell Biology, Metabolism & Cancer, Department of Biomolecular Health Sciences, Faculty of Veterinary Medicine, Utrecht University, Utrecht, The Netherlands. [3]Biomolecular Mass Spectrometry and Proteomics, Bijvoet Centre for Biomolecular Research, Utrecht University, Utrecht, The Netherlands. [4]Metabolics Group, Leiden University Medical Centre, Leiden, The Netherlands. [5]Human Nutrition and Health, Wageningen University, Wageningen, The Netherlands. [6]Institute of Clinical Chemistry and Pathobiochemistry, School of Medicine, Klinikum rechts der Isar, Technical University Munich, Munich, Germany. [7]TranslaTUM, Center for Translational Cancer Research, Technical University Munich, Munich, Germany. [8]Laboratory of Clinical Biochemistry and Metabolism, Department of General Paediatrics, Adolescent Medicine and Neonatology, Faculty of Medicine, Medical Centre - University of Freiburg, 79106 Freiburg, Germany. [9]Department of General Pediatrics, Adolescent Medicine and Neonatology, Faculty of Medicine, Medical Centre - University of Freiburg, 79106 Freiburg, Germany. ✉e-mail: Annemiek.vanspriel@radboudumc.nl

metabolism can provide cancer cells with significant growth advantages

Recent studies have implicated altered FA metabolism as a major oncogenic factor in aggressive B-cell lymphoma, including Burkitt and diffuse large B-cell lymphoma (DLBCL), the latter being the most common form of mature B-cell lymphoma[14–17]. Despite clear therapeutical advances over the past decades, aggressive B-cell lymphoma still accounted for 544,000 new cases and 260,000 mortalities in 2020[18]. Plateauing therapeutic response rates necessitate new insights into the biology of B-cell lymphoma. We have previously identified tetraspanin CD37 as an independent prognostic marker for patients with DLBCL[19]. Furthermore, CD37-deficiency leads to spontaneous development of aggressive B-cell lymphoma in mice[20]. Tetraspanins are four-transmembrane proteins that form nanoscale protein clusters in the plasma membrane[21]. These clusters are essential for cell proliferation, migration and signalling across cell types and tissues[22–24]. Aberrant tetraspanin expression has been associated with enhanced metastatic capacity, cell invasion, growth and survival in many cancers[25–27]. CD37 is predominantly expressed by B cells and is important for humoral immunity[28,29]. CD37 is involved in signalling of important metabolic modulators, including AKT and STAT3[20,30,31]. Although different studies have implicated lipid metabolism in aggressive B-cell lymphoma as a major oncogenic driver, molecular mechanisms underlying reprogramming towards lipid metabolism remain elusive. Here, we report tetraspanin CD37 as an essential and conserved inhibitor of fatty acid metabolism in aggressive B-cell lymphoma. CD37 interacts with fatty acid transporter protein 1 (FATP1) in the plasma membrane, and inhibits the uptake and processing of exogenous palmitate into energy and essential building blocks. Importantly, blocking FA metabolism of CD37-deficient B-cell lymphomas diminished proliferation and viability, potentiating therapeutical exploitation[32]. In conclusion, this work provides a molecular framework underlying metabolic reprogramming in aggressive B-cell lymphoma.

## Results

### Aggressive B-cell lymphomas use palmitate for energy production

We previously reported that ~60% of patients with aggressive DLBCL have aberrant CD37 expression in tumour tissues which is related to inferior clinical outcome[20,33]. To study the metabolic phenotype of these aggressive B-cell lymphomas, we generated CD37-deficient human DLBCL cell lines (BJAB, OciLy1) using CRISPR/Cas9 technology and used a CD37-deficient mouse model[20]. Oxygen consumption and glycolytic activity were quantified in lymphoma cells with (wild-type: WT) or without CD37 (CD37KO) using the Seahorse metabolic analyser. Glucose supplementation of lymphoma cells cultured under nutrient-limited conditions resulted in a significantly higher glycolytic response in WT cells, measured by extracellular acidification rate (ECAR) (Fig. 1a, b). Moreover, inhibition of mitochondrial energy production indicated a larger glycolytic reserve in WT lymphoma cells (Fig. 1c). Interestingly, CD37KO lymphoma cells did not increase their glycolytic activity in response to glucose when the alternative substrate was abundantly available, in contrast to WT cells (Fig. S1A–C). Together, these results indicate functional glycolytic machinery in both WT and CD37KO lymphoma, yet WT lymphomas have a higher glycolytic capacity. Next, we observed CD37KO lymphoma cells to be highly responsive to palmitate via increased mitochondrial oxygen consumption rate (OCR), in contrast to WT lymphoma cells (Fig. 1d, e). Mitochondrial uncoupling identified significantly increased spare respiratory capacity (SRC) in CD37KO lymphoma cells compared to WT cells, only when given access to palmitate (FA(16:0)) (Fig. 1f and S1D–F). Supplementation with either glucose or glutamine was insufficient to generate such a response or to facilitate metabolic flexibility (Fig. 1e, f). Similarly, supplementation with a different fatty acid, oleic

acid (FA(18:1)), failed to evoke a substantial mitochondrial response in CD37KO cells compared to WT cells (Fig. 1i and S1G–I). Quantification of ATP production confirmed the metabolic preference of the lymphoma cells. In concordance with the earlier observed increased glycolytic activity of WT lymphoma cells, WT cells produced significantly more energy from exogenous glucose relative to untreated cells, compared to CD37KO lymphoma cells (Fig. 1b, g). Moreover, inhibition of glycolysis with 2-deoxy-d-glucose (2-DG) had a bigger effect on WT lymphoma cells compared to CD37KO cells (Fig. S1J). CD37KO lymphoma cells generated significantly more energy compared to WT lymphoma cells when supplemented with palmitate, in line with the observed respiratory response to palmitate (Fig. 1h and S1K). Interestingly, deletion of another related tetraspanin, CD53, did not alter the metabolic phenotype of the lymphoma cells (Fig. S1I). Together, these results show that CD37-positive lymphoma cells are limited to a glycolytic response, whereas CD37-negative lymphoma cells drive mitochondrial fatty acid oxidation (FAO) of palmitate, leading to enhanced energy production (ATP).

### CD37-deficient lymphomas deplete palmitate from serum

We next investigated whether these different metabolic preferences in the lymphoma cell models could be recapitulated in vivo. To this end, serum was collected from healthy and tumour-bearing CD37KO mice and WT littermate controls and analysed for individual lipid classes[34]. CD37KO mice develop spontaneous aggressive B-cell lymphoma[20], in contrast to WT mice that only develop lymphoma in ~10% of the cases. Quantification of the concentration of different lipid species revealed no differences between healthy WT and CD37KO serum (Fig. 2a). In contrast, total levels of free fatty acids (FFA) were significantly lower in serum of tumour-bearing CD37KO mice compared to the serum of tumour-bearing WT mice (Fig. 2a). This was not a consequence of different growth conditions between WT and CD37KO mice (Fig. S2A) that received free chow and showed similar feeding behaviour. Other lipid species were not present in different quantities between WT and CD37KO serum of tumour-bearing mice, indicating that the depletion of FFA was specific. Further specification of the FFA species revealed that long-chain fatty acids (LCFAs) were depleted from serum of CD37KO mice, both in healthy and in tumour-bearing mice (Fig. 2b). Interestingly, LCFAs were significantly more depleted from serum of tumour-bearing CD37KO mice compared to healthy CD37KO mice, whereas the opposite was the case for WT tumour-bearing mice (Fig. 2b). Analysis of these LCFAs showed multiple different LCFA species to be depleted from CD37KO serum relative to WT serum (Fig. 2c). Notably, FA(16:0) and FA(18:0–2) were significantly lower in CD37KO serum of tumour-bearing mice compared to WT tumour-bearing mice. Interestingly, only palmitic acid (FA(16:0)), oleic acid (FA(18:1)) and linoleic acid (FA(18:2)) were significantly depleted from healthy CD37KO serum, relative to healthy WT serum (Fig. 2c). Moreover, FA(16:0) was the only FFA that was depleted from the serum of CD37KO mice in a tumour-dependent manner (Fig. 2c). Taken together, these data indicate that palmitate is systemically depleted in mice with lymphoma in a CD37-dependent manner.

### Premalignant CD37-deficient B cells have enhanced palmitate uptake potential and actively engage in fatty acid oxidation

As CD37 is also expressed in non-B cells and circulating palmitate levels are affected by different tissues (liver/fat), we performed a lipid analysis of purified WT and CD37KO primary B cells to study whether the phenotype was indeed B cell-dependent. No differences were found in lipid species composition between purified WT and CD37KO B cells (Fig. S2B). Furthermore, FFA composition was equal between WT and CD37KO B cells (Fig. S2C). The intracellular abundance of a metabolite at any given time is determined by the ratio between uptake and breakdown. We observed that the uptake potential of palmitate was significantly enhanced in CD37KO primary B cells compared to WT B

cells (Fig. 2d). Furthermore, ATP quantification demonstrated a significantly enhanced capacity of CD37KO primary B cells to process palmitate into energy compared to WT B cells (Fig. 2e). Contrastingly,

uptake of glucose was decreased in CD37KO primary B cells compared to WT cells (Fig. 2f). Since binding of carnitines to LCFA is required for functional mitochondrial processing and increased LCFA-carnitine

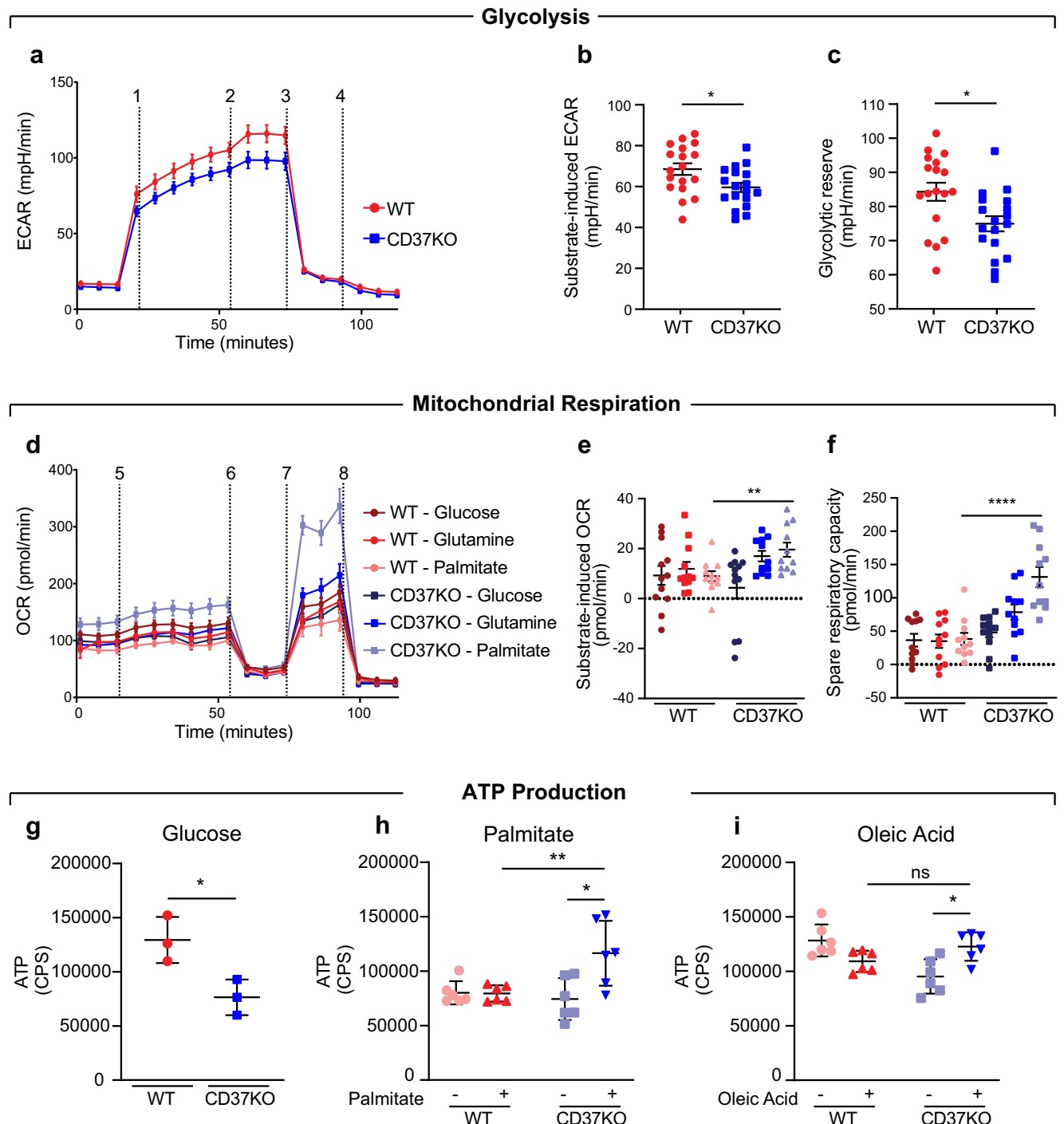

**Fig. 1 | Different metabolic preferences of WT and CD37KO human B-cell lymphomas.** Seahorse analysis of glycolysis (**a**–**c**) and mitochondrial respiration (**d**–**f**) in B-cell lymphoma. Cells (WT and CD37KO BJAB) were incubated in nutrient-restricted (**a**) or nutrient-rich (**d**) medium for 2 h and subjected to an acute substrate injection (1 in **a**, 5 in **d**) of glucose (10 mM), glutamine (10 mM), or palmitate (50 μM). Continuous extracellular acidification (ECAR) (**a**) or oxygen consumption ratio (OCR) (**d**) values are shown in response to 2, 6: Oligomycin A (1 μM), 3: 2-DG (20 mM), 7: FCCP (1 μM) and 4,8: Rotenone/AntimycinA (Rot/AA) (1 μM). Substrate-induced ECAR (n = 18, p = 0.0191) (**b**) or OCR (n = 12, p = 0.0032) (**e**) were calculated as the difference between baseline and acute substrate injection (1, 5). Glycolytic reserve (**c**) was calculated as the difference in ECAR between baseline and

Oligomycin A (2) (p = 0.0107). The spare respiratory capacity (**f**) was calculated as the difference in OCR between baseline and FCCP (7) (p < 0.0001). Relative ATP production (counts per second, CPS) was assessed in response to Glucose (n = 3, p = 0.0270) (**g**), Palmitate (n = 6, WT vs KO; p = 0.0145, KO− vs KO+; p = 0.0051) (**h**) and Oleic acid (n = 6, p = 0.0110) (**i**) after 2 h (**g**). Two-way unpaired t-test (**b**, **c**, **g**) or Two-way ANOVA with Tukey's post hoc test (**e**, **f**, **h**, **i**) were performed to check for significant differences between indicated groups, ns: not significant, *p < 0.05, **p < 0.01, ****p < 0.0001. Error bars represent mean ± SD. Experiments were repeated three times, yielding similar results. Source data are provided as a Source Data file. See also Supplementary Fig. 1.

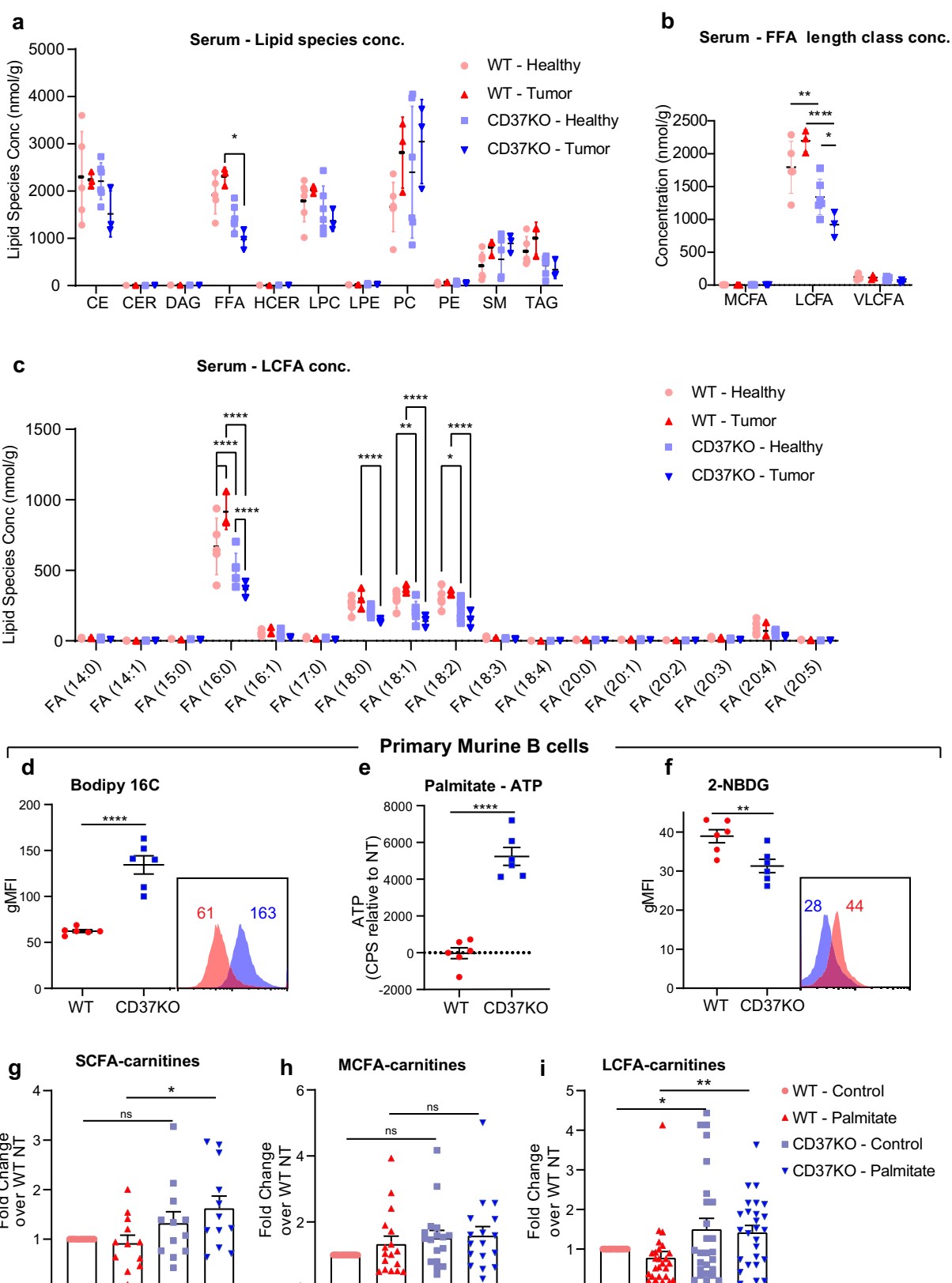

abundance is indicative of active uptake[35], we analysed different FA-carnitine species in WT and CD37KO B cells. Quantification of carnitine-FA species after palmitate supplementation showed palmitate-dependent accumulation of short-chain fatty acids (SCFAs)-carnitine abundance in CD37KO primary B cells (Fig. 2g), as well as increased LCFA-carnitines (Fig. 2i). Within the mitochondria, LCFAs are

detached from carnitine and subjected to fatty acid oxidation (FAO), to produce SCFAs and intermediates to fuel the TCA cycle[36]. Therefore, the observed accumulation of SCFA-carnitines in CD37KO B cells after palmitate supplementation signified the active breakdown of LCFAs in these cells. The lack of medium-chain fatty acid (MCFA)-carnitine accumulation confirmed that the accumulation of LCFA- and SCFA-

**Fig. 2 | Palmitate is selectively deprived of the serum of CD37KO lymphoma-bearing mice and is actively taken up and processed by CD37KO primary B cells.** Lipidomics analysis of serum of WT littermate control and CD37KO (healthy, $n = 6$ and lymphoma-bearing, $n = 3$) mice. Analysis of the mean abundance per lipid species; CE cholesterol esters, CER, HCER ceramides, DAG diacylglycerol, FFA free fatty acids ($p = 0.0491$), LPC lysophosphatidylcholine, LPE lysophosphatidylethanolamine, PC phosphatidylcholine, PE phosphatidylethanolamine, SM sphingomyelin, TAG triacylglyceride. **a** Absolute levels of FFAs were grouped and pooled by length-class (medium-chain fatty acids (MCFA), long-chain fatty acids (LCFA), and very long-chain fatty acids (VLCFA)) and quantified (LCFA: WT− vs KO−; $p = 0.0016$, WT+ vs KO+; $p < 0.0001$, KO− vs KO+; $p = 0.0213$) (**b**). Absolute concentrations of individual LCFA species were quantified (16:0; $p < 0.0001$, 18:0; $p < 0.0001$, 18:1; WT− vs KO−; $p = 0.0058$, WT+ vs KO+; $p < 0.0001$, 18:2; WT− vs KO−; $p = 0.0102$, WT+ vs KO+; $p < 0.0001$) (**c**). Freshly isolated CD43-, B220 + splenocytes from WT ($n = 6$) and CD37KO ($n = 6$) mice of 2–3 months old were stained with fluorescent analogues for palmitate (Bodipy FL C16) ($n = 6$, $p < 0.0001$) (**d**) and

glucose (2-NBDG) ($n = 6$, $p = 0.0095$) (**f**) to assess the uptake potential using flow cytometry. Relative ATP production (counts per second, CPS) compared to non-treated cells was assessed after 2 h of palmitate (50 μM) ($n = 6$, $p < 0.0001$) (**e**) supplementation. Short-chain fatty acid (**g**) (SCFA, $p = 0.0163$), medium-chain fatty acid (**h**) (MCFA) and long-chain fatty acid (**i**) (LCFA, control; $p = 0.0345$, palmitate; $p = 0.0038$)-carnitines in primary murine WT (Control: $n = 3$, Palmitate: $n = 3$) or CD37KO (Control: $n = 3$, Palmitate: $n = 3$) B cells were grouped per class and quantified with ultra-performance liquid chromatography system coupled to a tandem mass spectrometer after palmitate supplementation (50 μM). Two-way unpaired $t$-test (**d–f**) or two-way ANOVA with Tukey's post hoc test (**a–c**, **g–i**) were performed to check for significant differences between the indicated groups, ns not significant, $*p < 0.05$, $**p < 0.01$, $****p < 0.0001$. Error bars represent mean ± SD. Data in **a–f** was derived from two independent experiments, and experiments in **g–i** were repeated once, with biologically independent groups. Source data are provided as a Source Data file. See also Supplementary Fig. 2.

carnitines in CD37KO cells truly reflected active engagement in oxidative turnover and not a general increase in all FAs (Fig. 2h). Together, these results show that CD37-deficiency in mice leads to increased uptake and processing of palmitate by CD37KO B cells, confirming the earlier observed phenotype in human lymphoma B cells.

### CD37-deficient lymphoma cells actively process palmitate into essential building blocks

Next, we investigated whether the enhanced FAO in CD37KO cells (Figs. 1 and 2) was a direct consequence of exogenous palmitate using carbon tracing studies that also provide detailed insights into what kind of metabolites and building blocks are produced directly from palmitate. Palmitate is processed in the FAO pathway, generating multiple NADH moieties and acetyl-CoA during each cycle (full palmitate breakdown generates eight acetyl-CoA moieties that can feed into the TCA cycle). Upon palmitate supplementation, FAO-enzymes will compete between newly imported 16 C and already partially oxidised shorter acyl-moieties. As such, the accumulation of SCFAs is indicative of active mitochondrial uptake and processing of FAs[36,37]. WT and CD37KO lymphoma cells were supplemented with [13]C- palmitate to study to what extent these cells process exogenous palmitate directly. Since the two [13]C-carbons of [13]C-acetyl-CoA feed into the TCA cycle, quantifying to what extent metabolites contain at least two [13]C-isotopes provides information on whether cells use palmitate for energy production, or prioritise the use of TCA-intermediates in more anabolic processes (Fig. 3a). Interestingly, supplementation with [13]C-palmitate resulted in distinctively different total labelling profiles for WT or CD37KO lymphoma cells, meaning they prioritise different metabolic pathways (Fig. 3b). The sum of M + 0 plus all the labelled isotopes in SCFA-carnitines accumulated in CD37KO cells, confirming that the increase in FAO is a direct consequence of processing exogenous palmitate (Fig. 3c and S3A), in line with our mouse studies. Furthermore, increased accumulation of TCA cycle-intermediates such as citrate, succinate and malate in CD37-deficient lymphoma cells indicated that palmitate was degraded via FAO to fuel the TCA cycle via acetyl-CoA (Fig. 3d and S3B). This is in line with the observed increased palmitate-dependent SRC observed in CD37KO cells (Fig. 1d, f). In addition, the TCA cycle-associated amino acid synthesis profile, was completely different between WT and CD37KO lymphoma cells (Fig. 3e and S3C). Correlation-based clustering revealed that predominantly aspartic acid (ASP), asparagine (ASN) and glutamic acid (GLU) were different between WT and CD37KO cells (Fig. 3e and S3C). Statistical analysis of the different pathways revealed M(n + 2) labelled metabolites to be specifically accumulated in CD37KO over WT cells, independent of M + 0 isotope labelling, confirming specific incorporation of [13]C from exogenous palmitate. Incorporation of [13]C from palmitate into SCFA-FAO intermediate [13]C-4.0-Butyryl-L-carnitine (Fig. 3F) and TCA intermediates [13]C-citrate (Fig. 3g) and [13]C-malate (Fig. 3h)

occurred significantly more in CD37KO lymphoma cells than in WT cells. These intermediates can either be used for energy production or for building blocks such as membrane lipids (via citrate) and amino acids (Fig. 3a)[9,38]. Interestingly, the essential metabolites for nucleotide synthesis[16,39], amino acids [13]C-asparagine and [13]C-aspartic acid (Fig. S3I), accumulated significantly in CD37KO cells after [13]C-palmitate supplementation compared to WT cells, especially after 8 h (Fig. 3i). Together, these results show that lymphoma cells lacking CD37 actively process exogenous palmitate via FAO into TCA intermediates and essential building blocks (such as citrate and ASP), confirming a CD37-dependent metabolic shift in these tumour cells.

### CD37 inhibits palmitate uptake in B-cell lymphoma via interaction with FATP1

Next, we investigated the molecular mechanism underlying CD37-dependent inhibition of uptake and processing of exogenous palmitate in aggressive lymphoma. We first established the uptake potential of palmitate in human lymphoma cells, since palmitate was specifically deprived from serum (Fig. 2) and was actively processed in B cells of CD37-deficient mice (Fig. 2) and human lymphoma cells (Fig. 1). Indeed, uptake of fluorescent palmitate analogue (Bodipy FL C16) was significantly enhanced in CD37KO lymphoma cells compared to WT lymphoma cells (Fig. 4a). To verify that this distinctive uptake profile was a direct consequence of CD37 expression, we performed rescue experiments in which full-length CD37 was re-introduced in CD37KO lymphoma cells. CD37 re-expression in CD37KO lymphoma cells resulted in a significant reduction in uptake of palmitate compared to mock-transfected cells, resembling the WT phenotype (Fig. 4b). This was specific for palmitate, since lauric acid fluorescent analogue (Bodipy 12 C) and glucose analogue uptake (fluorescent tracer, 2-NBDG) did not differ between these genotypes (Fig. 4c, d). Furthermore, CD37KO lymphomas increased their uptake of glucose upon re-introduction of CD37 (Fig. 4e). To exclude that these findings were restricted to one lymphoma cell line, we analysed several different CD37-positive and CD37-negative lymphoma cell lines for their response to palmitate. Indeed, endogenous CD37-negative lymphoma cell lines (OciLy19 and SUDHL6) resembled the CD37KO phenotype and displayed increased ATP producing capacity and spare respiratory capacity (SRC) upon palmitate supplementation, which was abolished by an FA-metabolism inhibitor, etomoxir (Fig. S4A–I). In contrast, this palmitate-dependent metabolic response was absent in endogenously CD37-positive lymphoma cell lines (OciLy8, DOHH2) (Fig. S4E–J), resembling the WT phenotype (Fig. S4A, B). Collectively, these results support a universal role for CD37 in the direct inhibition of LCFA uptake in B-cell lymphoma.

Uptake of LCFAs occurs via different fatty acid transporters (Solute Carrier Family 27 members (*Slc27A*))[40] and scavenging receptors (CD36) on the cell membrane[41]. RNA-sequencing analysis of these

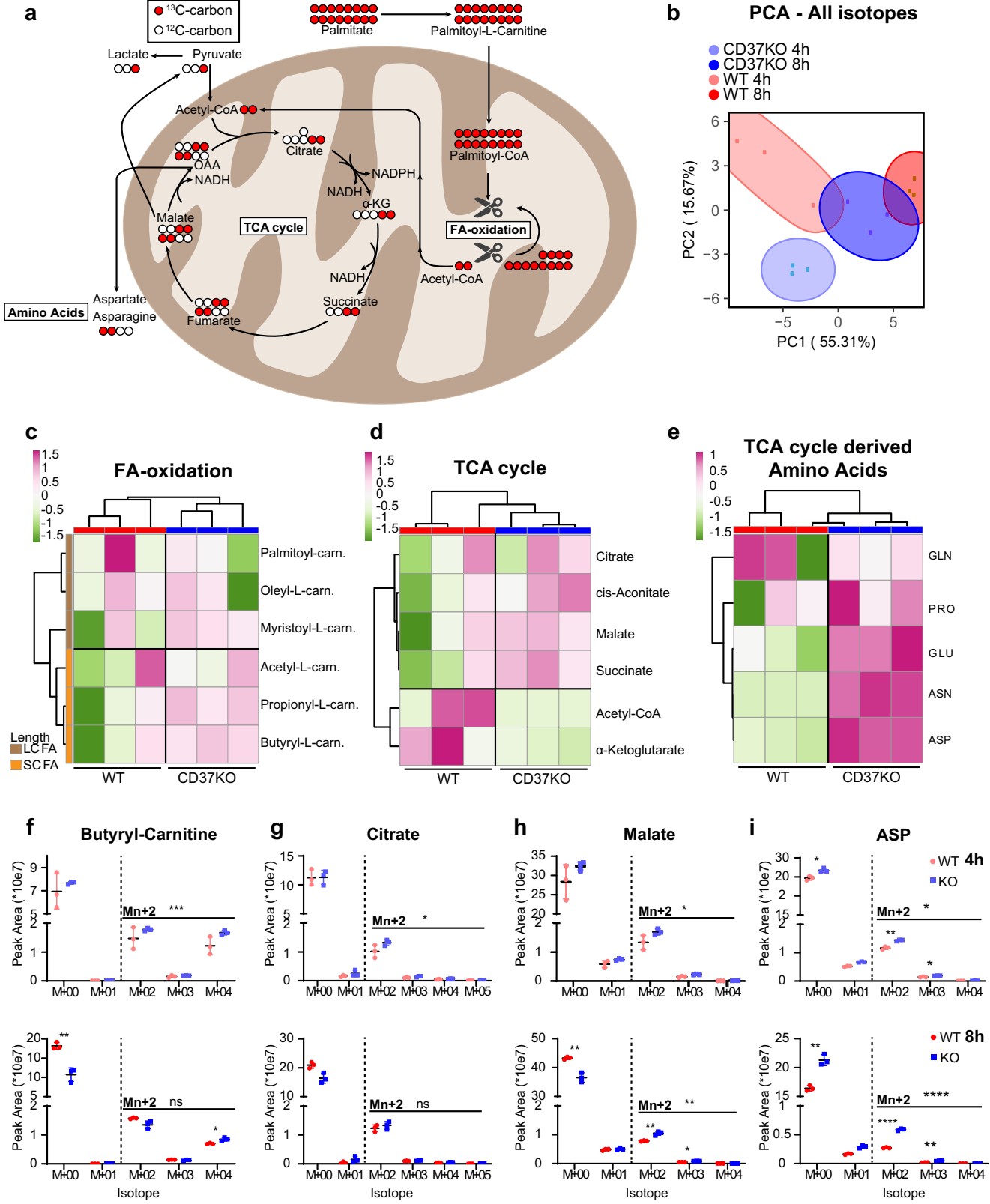

fatty acid transporters in primary B cells revealed prominent expression of fatty acid transporter protein 1 (*Slc27a1*, FATP1) and of CD36 (Fig. S4M). Quantification of FATP1 and CD36 protein expression on the surface of WT and CD37KO lymphoma cells revealed no difference between these cell genotypes (Fig. S4N, O). However, dual labelling of CD37 and the palmitate transporters revealed obvious colocalization between CD37 and FATP1 (Pearson's $R = 0.71$), but significantly less

between CD37 and CD36 (Pearson's $R = 0.41$) (Fig. 4f, g). Next, we investigated whether CD37 could inhibit palmitate uptake via protein interaction with FATP1 in the plasma membrane using proximity ligation assays (PLA) that allows for in situ detection of endogenous protein-protein interactions within nanometre distance[42]. A specific signal indicative of direct interaction between CD37 and FATP1, was observed in WT cells (Figs. 4h, I and S4P). Furthermore, FATP1 was

**Fig. 3 | ¹³C-palmitate carbons are incorporated into TCA intermediates and essential building blocks in CD37KO lymphoma cells.** WT ($n = 3$) and CD37KO ($n = 3$) lymphoma (BJAB) cells were supplemented with ¹³C-palmitate (50 μM) for 4 and 8 h. Each of the ¹³C-palmitate carbons (red circles) can be incorporated into different metabolic pathways (**a**). Principle component analysis (PCA) revealed group defining, differential incorporation between WT and CD37KO after 4 or 8 h of ¹³C-palmitate supplementation (**b**). Levels of M + 0 plus labelled M + n ¹³C-long-chain fatty acid-carnitines and ¹³C-short-chain fatty acids (**c**), ¹³C-TCA intermediates (**d**), and ¹³C-amino acids (**e**) after 4 or 8 h of ¹³C-palmitate supplementation were Log-transformed and analysed in greater detail with correlation-based-clustering. Sums of peak areas of all isotopes of significant values from the different pathways in **c–e** after 4 and 8 h were quantified (**f** -8 h M + 00; $p = 0.0027$, M + 04; $p = 0.0170$, **h** -8 h M + 00; $p = 0.0097$, M + 02; $p = 0.0024$, **i** -4 h M + 00; $p = 0.0215$, M + −2; $p = 0.0065$, M + 03; $p = 0.0208$, **f** -8 h M + 00; $p = 0.0037$, M + 02; $p < 0.0001$, M + 03; $p = 0.0036$) (**f–i**). Sums of M + 2 labelling patterns were separately analysed (Mn + 2, **f** -4 h; $p = 0.0007$, **g** -4 h; $p = 0.0188$, **h** -4 h; $p = 0.0017$, **h** -8 h; $p = 0.0071$, i -4 h; $p = 0.0132$, **i** -8 h; $p < 0.0001$). Two-way ANOVA with Tukey's post hoc test of **c–e** were performed to check for significant differences between the indicated groups (**f–i**), ns not significant, *$p < 0.05$, **$p < 0.01$, ***$p < 0.001$, ****$p < 0.0001$. Error bars represent mean ± SD. Data were derived from six CD37KO and six CD37-positive (WT) independent human lymphoma cell cultures in one experiment. Source data are provided as a Source Data file. See also Supplementary Fig. 3.

co-immunoprecipitated with CD37 upon overexpression of FATP1-FLAG and CD37-alphaTAG in CD37KO lymphoma cells (Fig. 4j, red: 63 kDa). As a control, overexpression with an empty vector did not result in a positive signal when stained against FLAG-TAG, and transfection efficiency of CD37 was equal between control and FATP1 transfected cells (Fig. 4k: 30–45 kDa). Together, these results demonstrate a specific interaction between tetraspanin CD37 and the LCFA-transporter FATP1 on the cell surface of aggressive B-cell lymphoma.

## Inhibition of FATP1 reversed the CD37-dependent fatty acid metabolic switch

Since CD37 has different interaction partners[20,28], some of which can affect signalling, we next studied whether the CD37-dependent metabolic switch was caused by loss of FATP1 inhibition. To assess this question, we treated CD37KO and WT lymphoma cells with specific inhibitors for FATP1, arylpiperazine 5k (DS22420314, referred to as compound "5k" and "12a")[43,44]. First, we verified that the FATP1 inhibitors, did not affect the viability of the WT and CD37KO lymphoma cells directly (Fig. S5A). In the presence of compound 5k, the palmitate-dependent substrate-induced OCR and the SRC of CD37KO lymphoma cells was abolished and mostly restored to WT levels (Fig. 5a–c). Interestingly, the inhibition was specific for palmitate, since there was no 5k-dependent effect observed on lymphoma B cells supplemented with oleic acid (FA(18:1)) (Fig. 5d–f). Furthermore, ATP quantification after FATP1 inhibition in a full growth medium, supplemented with palmitate, decreased the total amount of ATP produced in CD37KO lymphoma cells, whereas WT cells treated with compound 5k showed no difference in energy production (Figs. 5g and S5C). Moreover, the combination of 5k and palmitate decreased ATP accumulation relative to palmitate only in an endogenously CD37-negative cell line (OciLy19), in contrast to a CD37-positive cell line (OciLy8), which did not display a decrease in palmitate-dependent ATP production upon FATP1 inhibition (Fig. S5B). Additionally, uptake of palmitate fluorescent analogue decreased in a concentration-dependent manner of 5k and 12a in both WT and CD37KO lymphoma cells, yet the effect was bigger in CD37KO (Fig. 5h, i) and in endogenously CD37-negative lymphoma cells (Fig. S5D). This concentration-dependent inhibition of palmitate uptake was confirmed in primary CD37-deficient B cells (Fig. S5E, F). In contrast, uptake of lauric acid (an MCFA, not dependent on FATP1) was completely unaffected by compounds 5k and 12a, in both lymphoma cell lines and primary B cells (Fig. S5F–H). Together, these results confirm that FATP1 is the primary transporter responsible for the fatty acid metabolic switch in the absence of CD37.

## Inhibition of fatty acid metabolism detrimental to CD37KO lymphoma

Next, we investigated whether the CD37-dependent metabolic shift towards FA metabolism due to lack of FATP1 inhibition could be therapeutically targeted. WT and CD37KO lymphoma cells were treated with an inhibitor of the mitochondrial enzyme carnitine-palmitoyl-transferase 1a (CPT1a). CPT1a catalyses the binding of an activated long-chain fatty acid to carnitine resulting in a long-chain acylcarnitine that can pass the mitochondrial membrane. Therefore, CPT1a is one of

the key enzymes responsible for the transport of LCFAs, including palmitate, into the mitochondria[45]. Strikingly, inhibition of CPT1a with ST1326 resulted in significantly lower viability and proliferation of CD37KO lymphoma compared to WT lymphoma cells in a concentration-dependent manner (Figs. 6a–c and S6A–D). Furthermore, palmitate processing into energy was abolished in CD37KO lymphoma cells treated with CPT1a inhibitor in a concentration-dependent manner (Fig. 6d). Additionally, the palmitate-dependent increased spare respiratory capacity (SRC) of endogenous CD37-negative (OciLy19, SUDHL6) lymphoma cell lines was lost when treated with the CPT1a inhibitor (Fig. S6K–N), whereas no effect of palmitate supplementation in combination with CPT1a inhibition was observed in endogenous CD37-positive (OciLy8, WSU-NHL) lymphoma cells (Fig. S6G–J). Since palmitate is delivered to the mitochondria from both exogenous sources and endogenous lipolysis (lipid droplets) during catabolic states in vivo, we assessed whether exogenous sources were responsible for the viability and proliferation of CD37KO lymphoma by inhibiting long-chain-fatty acid-CoA ligases (ACSLs). ACSLs catalyse the activation of exogeneous FFAs and, as such are essential for the effective processing of exogenous palmitate[46]. The viability of both WT and CD37KO lymphoma cells decreased in a concentration-dependent manner (Figs. 6e, f and S6D–F), yet CD37KO cells were significantly more affected than WT cells, indicating more dependency of CD37KO lymphomas on fatty acid activation. Interestingly, whereas CPT1a inhibition in CD37KO cells mostly resulted in apoptosis (Fig. S6A, B), ACSL1 inhibition resulted in more necrosis (Fig. S6F). Similar to CPT1a inhibition, palmitate-induced energy production by CD37KO lymphoma cells was significantly diminished upon treatment with ACSL1 inhibitor compared to WT control cells (Fig. 6g). Importantly, these cells were cultured in nutrient-rich conditions, replete with alternative energy substrates, yet this was not sufficient for CD37KO lymphoma cells to overcome the limitations posed by inhibition of FA metabolism. Collectively, these results indicate that CD37KO lymphoma cells become dependent on the FA-metabolic switch and, as such, can be targeted by metabolic inhibitors to decrease survival and proliferation.

## Primary human CD37-negative lymphomas have enhanced lipid storage capacity

Besides direct processing of palmitate via FAO, some cells can store fatty acids into specialised organelles called lipid droplets (LDs). These LDs have been correlated to chemoresistance and worse prognosis of patients[3,47–49]. To investigate whether CD37-deficiency affected lipid storage, WT and CD37KO lymphoma cells were analysed for the presence of lipid droplets. In unstimulated cells, the quantity of lipid droplets was significantly higher in CD37KO lymphoma cells compared to WT lymphoma cells (Figs. 7a and S7A). Furthermore, supplementation of palmitate led to a dose-dependent increase in LDs in CD37KO lymphoma cells, whereas WT cells did this to a far lesser extent (Fig. 7b). Interestingly, higher concentrations of palmitate led to diminished viability of WT lymphoma cells, potentially due to their inability to form new LDs and subsequent lipotoxicity. In contrast, CD37KO cells remained viable even with the highest concentration of

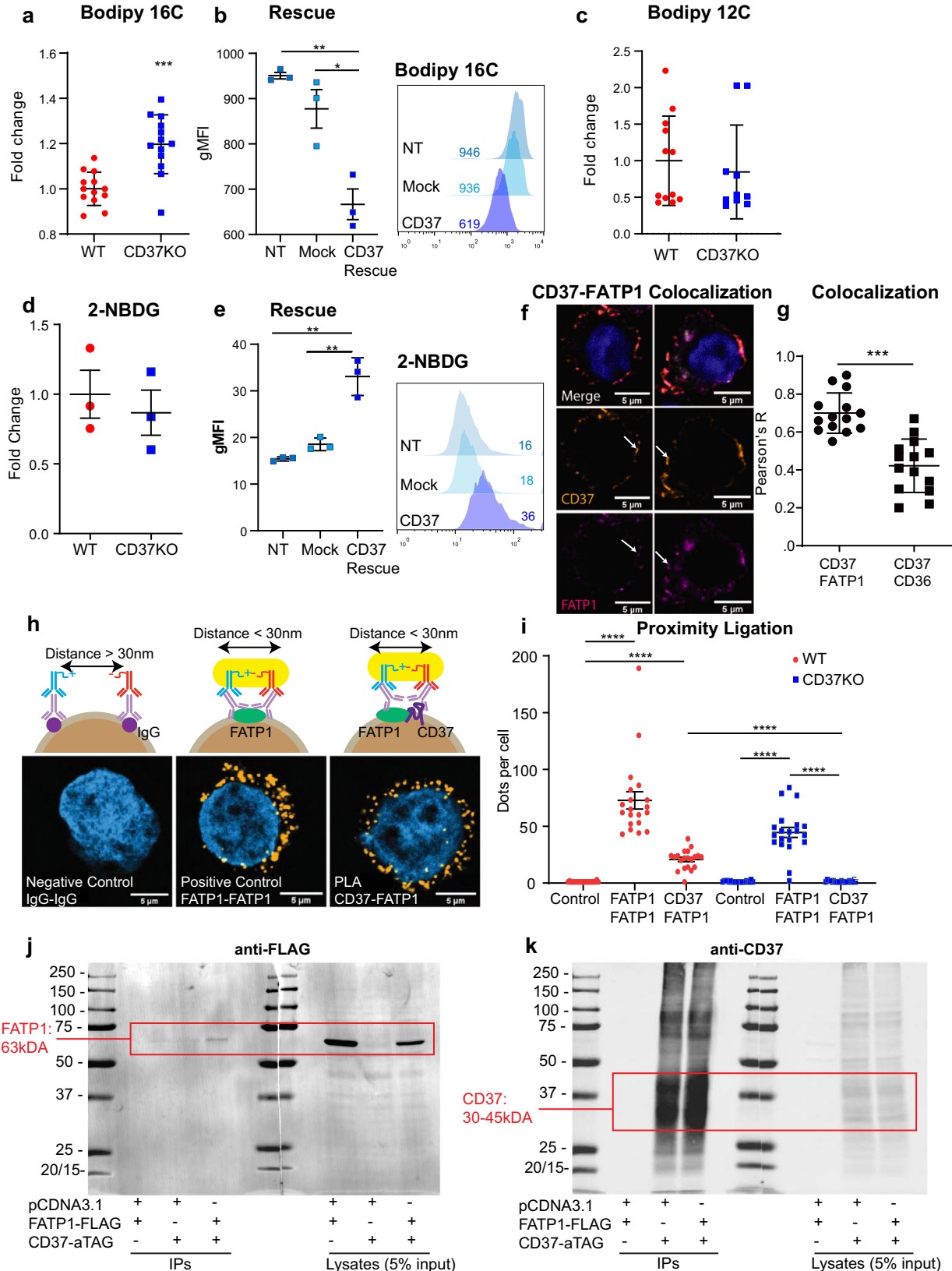

palmitate (Fig. S7B). Moreover, inhibition of fatty acid synthase (FAS) resulted in significantly more necrotic CD37KO lymphoma cells (Fig. S7C, D). Furthermore, inhibition of FAS did result in a significant reduction in ATP availability only in CD37KO lymphoma cells (Fig. S7E). FAS is an essential enzyme involved in lipid storage in LDs[50] that has been reported to be overexpressed in non-Hodgkin lymphoma[51],

mantle cell lymphoma[52], and other cancer types[10]. Overall, these results indicate an enhanced capacity of CD37KO lymphoma cells to survive and store exogenous FAs in lipid droplets, which is essential for long-term energy supply and prevention of lipotoxicity.

Finally, we investigated whether the enhanced storage capacity of CD37-deficient lymphoma cells was also observed in primary human

**Fig. 4 | CD37 inhibits the uptake of palmitate in B-cell lymphoma via interaction with fatty acid transporter FATP1.** Human lymphoma BJAB cells supplemented with fluorescent analogue for palmitate (Bodipy FL C16) ($n = 13$, $p = 0.00010$) (**a**), lauric acid analogue (Bodipy FL C12) ($n = 10$) (**c**) and glucose analogue (2-NBDG) ($n = 3$) (**d**) to assess uptake potential with flow cytometry. BJAB CD37KO cells ($n = 3$) were transfected or mock transfected with a CD37 construct and stained with Bodipy FL C16 (**b**) or 2-NBDG (**e**) to check uptake potential after CD37 rescue (**b**: NT vs rescue; $p = 0.0012$, NT vs Mock; $p = 0.0178$, **e** NT vs rescue; $p = 0.0017$, NT vs Mock; $p = 0.0042$). Confocal microscopic analysis revealed colocalization of CD37 (**f**, yellow, middle panels) and FATP1 (**f**, magenta, bottom panels) in the membranes of WT human lymphoma cells ($n = 13$) (blue: DAPI). Pearson's R ($p < 0.0001$) analysis indicates the likelihood of CD37 and FATP1, or CD37 and CD36, to reside in the same microdomain (**g**). Visualisation of in situ proximity ligation on cells stained with isotype controls ($n = 21$) (**h**, left panel), positive control ($n = 21$) (two different primary antibodies against FATP1) (**h**, middle panel) and CD37 and FATP1 ($n = 20$) (**h**, right panel). The dots per cell were quantified over three independent experiments and visualised for one representative repeat (all; $p < 0.0001$) (**i**) CD37KO lymphoma cells (BJAB) were transfected with empty vector or FATP1-FLAG construct and alfa-tagged (aTAG) CD37 construct and subjected to pulldown on alfa-tagged CD37. Blots were stained with anti-FLAG antibody to reveal specific interaction between CD37 and FATP1 (FATP1: 63 kDa, red square) (**j**), and anti-CD37 antibody (30–45 kDa (glycosylation)) (**k**). Two-way unpaired $t$-tests (**a**, **c**, **d**, **g**) or Two-way ANOVA with Tukey's post hoc test (**b**, **e**, **i**) were performed to check for significant differences between the indicated groups, $^{*}p < 0.05$, $^{**}p < 0.01$, $^{****}p < 0.0001$. Error bars represent mean ± SEM. Data in **a**–**e** and **g** was derived from three independent experiments, **h**–**k** were repeated at least three times, yielding similar results. gMFI geometric mean fluorescence intensity. Source data are provided as a Source Data file. See also Supplementary Fig. 4.

B-cell lymphoma tissues. To this end, a panel of diffuse large B-cell lymphoma (DLBCL) tissues ($n = 22$) were analysed for the presence of lipid deposits using oil red O (ORO) staining. The panel consisted of 15 CD37-positive DLBCLs and seven CD37-negative DLBCLs, the latter correlating to inferior clinical outcome[20]. Large lipid deposits were found in 100% of CD37-negative DLBCL ($n = 7$) (Fig. 7c, d), in contrast to only 27% of CD37-positive DLBCLs cases ($n = 15$) (Fig. 7e, f). In addition, CD37-negative DLBCLs contained a larger ORO-positive tissue area compared to CD37-positive, ORO-positive DLBCLs. These large macro-structured lipid droplets correlated with more intracellular LDs per lymphoma cell in CD37-negative DLBCL tissue (Fig. 7g, h). Cells residing in close proximity to the large lipid deposits were often observed full of intracellular lipid droplets (Fig. 7g, panel 1). Furthermore, the average LD in CD37-negative DLBCL tissues was larger compared to CD37-positive DLBCL tissues (Fig. 7i). Finally, we analysed expression levels of FA-metabolism associated genes in CD37-negative and positive primary B cells and found phospholipase 2g2d (*Pla2g2d*) as most significantly upregulated in CD37-negative B cells (Fig. S7F). PLA2g2d can be secreted to release fatty acids from complex extracellular lipid structures, such as macro-lipid deposits[53]. Indeed, database analysis revealed *PLA2g2d* to be highly expressed by DLBCL tissues compared to paired-healthy tissues, whereas this was not the case for any other form of cancer (Fig. S7G, H). In summary, these findings demonstrate that patients with CD37-deficient B-cell lymphoma accumulate lipids for storage within tumour tissues.

## Discussion

Aggressive lymphomas are often associated with altered metabolic phenotypes[16,17,54–57]. Besides the metabolic switch from oxidative respiration to aerobic glycolysis, the importance of other metabolic pathways has recently been established. Lipid metabolism has gained particular attention because changes in this pathway have been found across many different tumour types[4–10]. Still, the exact molecular mechanisms underlying these metabolic alterations and their relation to the inferior clinical outcome of patients have remained largely elusive. Our work identified tetraspanin CD37 as an essential membrane-bound gatekeeper for a fatty acid metabolic switch in aggressive B-cell lymphoma via inhibitory interaction with FATP1.

While previous studies focused predominantly on deciphering protein expression and function of metabolic enzymes to explain altered metabolic phenotypes of cancers, our study demonstrates that fatty acid metabolism is already orchestrated upstream, at the level of the plasma membrane. The biology of fatty acid transporter FATP1 at the molecular level is not well understood, but knockout studies have shown that FATP1 is required for the transport of LCFA and subsequent fatty acid oxidation in various tissues[58]. There are indications that oligomerization of FATP1 at the cell surface enhances its transporter function[59], something that CD37 could potentially prevent. Irrespective of the type of molecular interaction, our data demonstrate that

CD37 is a key modulator of fatty acid metabolism via FATP1 inhibition in B-cell lymphoma.

Aberrant expression of tetraspanins has been reported in a variety of cancer types and affects cell adhesion, migration and metastasis through protein interactions with adhesion receptors, growth factor receptors and signalling molecules[26]. Specific tetraspanin interactions with transporter proteins have only recently been reported, including CD9 with glutamine transporter ASCT2[60] and TM4SF5 interaction with the amino acid transporter SLC38A9[61]. These studies demonstrate the importance of tetraspanin-partner interactions for the functional regulation of (metabolic) transporters. Tetraspanin CD37 is of special interest due to its prognostic effect on clinical outcomes for lymphoma patients. Aberrant CD37 expression is common in aggressive B-cell lymphoma, exemplified by 60% of patients with CD37-negative DLBCL who have inferior clinical outcome[19,20], and similar findings have been recently reported for patients with follicular lymphoma[62]. The altered fatty acid metabolism of CD37-negative lymphoma is beneficial for these tumours in multiple ways. Exogenous and endogenous palmitate provides CD37-negative lymphoma cells with high amounts of substrate for breakdown via FAO to fuel the TCA cycle and high production of energy. Furthermore, the superior influx of acetyl-CoA via FAO allows excessive TCA intermediates to be used for alternative, more anabolic processes in CD37-deficient lymphoma. These processes include lipid synthesis via citrate and synthesis of amino acids such as asparagine (ASN) and aspartic acid (ASP) for nucleotide synthesis. Interestingly, both citrate[63,64] and ASP[38] have gained a lot of attraction in recent years due to their rate-limiting effect on tumour cell proliferation, and inhibition of the ASP-ASN metabolic pathway in acute lymphoblastic leukaemia (ALL) is the first example of a tumour-metabolism-targeting therapy[65].

In addition to this beneficial phenotype for proliferation, aberrant lipid metabolism has been associated with increased chemoresistance in breast cancer[66–68], ovarian cancer[69–71], pancreatic cancer[72,73], head and neck cancer[74], and myeloid leukaemia[75]. Our data is in line with recent gene expression profiling that revealed specific upregulation of FATP1 (and not other fatty acid transporter proteins) during the progression of indolent lymphoma into aggressive DLBCL, further supporting that FATP1-dependent metabolic pathway are key in tumour advancement[76]. Furthermore, increased expression of FATP1 is correlated with decreased overall survival in patients with breast cancer[44]. We now show that loss of CD37 reprograms lymphoma cells into high-energy producing, pro-survival/pro-proliferating aggressive tumours. This reprogramming is dependent on enhanced fatty acid metabolism via FATP1, which may underlie the low response rate to chemotherapy of patients with CD37-negative lymphoma[19].

Whether CD37 also controls FA metabolism in healthy B cells (and other immune cells) remains unknown. Our finding that the FA profile is altered already in premalignant B cells upon loss of CD37 indicates an important physiological role of CD37 in the lipid metabolism of healthy cells. B-cell function and antibody

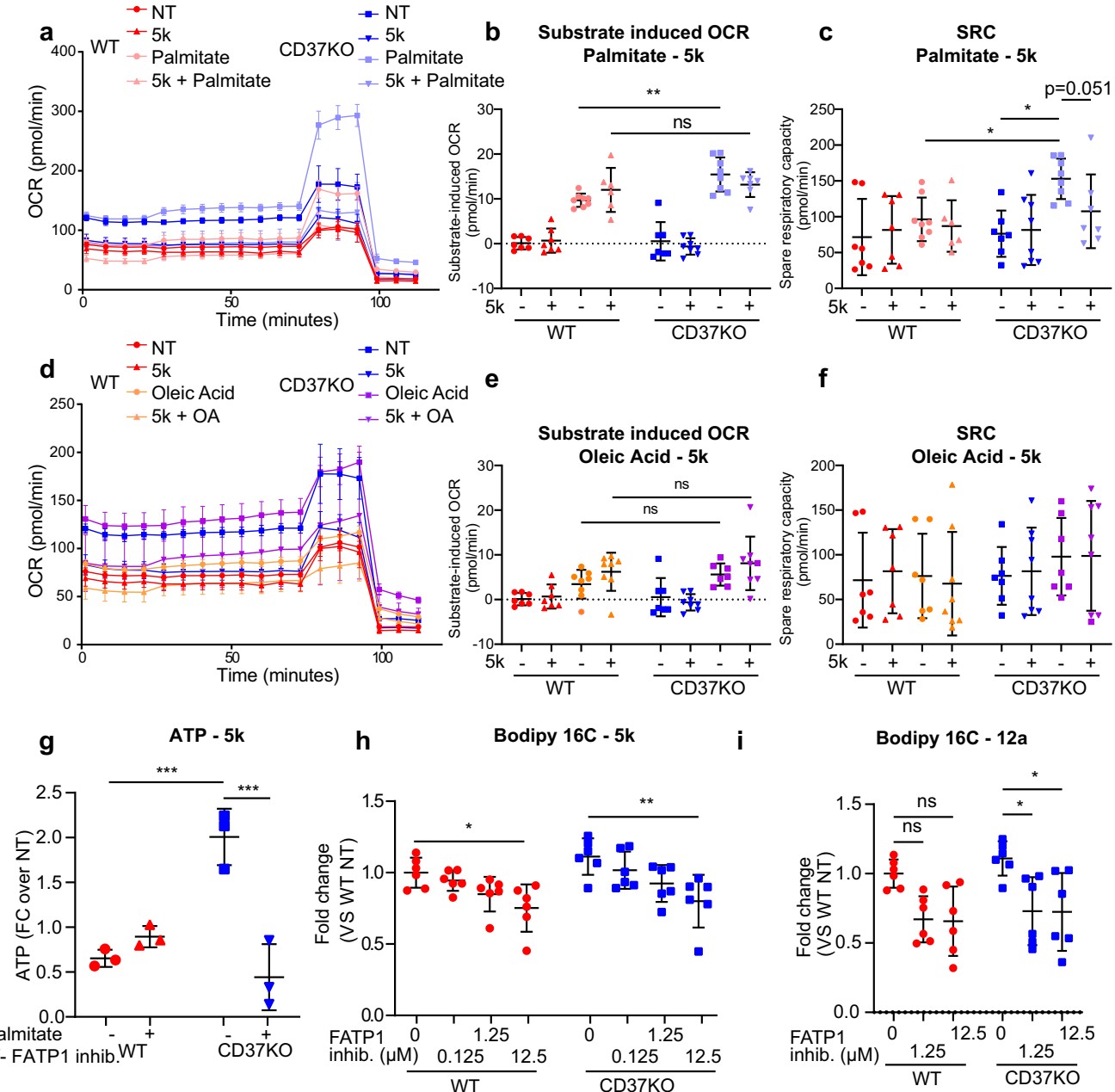

**Fig. 5 | Inhibition of fatty acid transporter FATP1 abolishes fatty acid metabolic switch in CD37KO B-cell lymphoma.** WT and CD37KO lymphoma cells (BJAB) were incubated in a nutrient-rich medium with or without FATP1 inhibitor (compound 5k (12.5 μM)) for 2 h. Cells were then transferred to a nutrient-restricted medium with ($n = 8$) or without ($n = 8$) compound 5k and subjected to an acute substrate injection of medium or palmitate (50 μM) (**a**) or oleic acid (50 μM) (**d**) at $t = 30$ min. Continuous oxygen consumption ratio (OCR) values are shown in response to FCCP (1 μM) at $t = 70$ min and Rotenone/AntimycinA (Rot/AA) (1 μM) at $t = 100$ min. Substrate-induced OCR (**b**, **e**) were calculated as the difference between baseline and acute substrate injection (**b**: $p = 0.0099$). The spare respiratory capacity (SRC) (**c**, **f**) was calculated as the difference in OCR between baseline and FCCP (**c**: WT vs KO; $p = 0.0365$, KO vs KO palmitate; $p = 0.0188$). Relative ATP production (counts per second, CPS) was assessed in response to 24-h FATP1 inhibition with compound 5k (12.5 μM) ($n = 3$, WT− vs KO−; $p = 0.0008$, KO− vs KO+; $p = 0.0003$) (**g**). Decrease in palmitate uptake was measured after 2 h FATP1 inhibition with compound 5k ($n = 6$, WT; $p = 0.0147$, KO; $p = 0.0033$) (**h**) and 12a ($n = 6$, KO; $p = 0.0355$, $p = 0.0315$) (**i**) and compared to non-treated cells ($n = 6$). Experiments were repeated three times, yielding similar results. Two-way ANOVA with Tukey's post hoc test were performed to check for significant differences between the indicated groups, ns not significant *$p < 0.05$, **$p < 0.01$, ***$p < 0.001$ ****$p < 0.0001$. Error bars represent mean ± SD. Source data are provided as a Source Data file. See also Supplementary Fig. 5.

production is impaired in CD37-deficient mice[28], although the role of altered lipid metabolism in this immune phenotype needs to be investigated. The expression profile of CD37 differs throughout B-cell development with a low expression on pre-B cells, high on mature B cells and absent on plasma cells[77], which may be linked to the importance of lipid metabolism in healthy or malignant plasma cell functioning[78,79]. A small subset of the healthy B cells

that participate in the germinal centre response, becomes glycolytic in response to serine/threonine protein kinase (GSK3) inhibition and cMyc accumulation[80]. Although this suggests a survival benefit for WT B cells over CD37KO B cells since the former is more glycolytic, maintenance of high levels of TCA intermediates increases the competitive fitness of some aggressive lymphomas in germinal centres[81]. Moreover, the lipid droplet

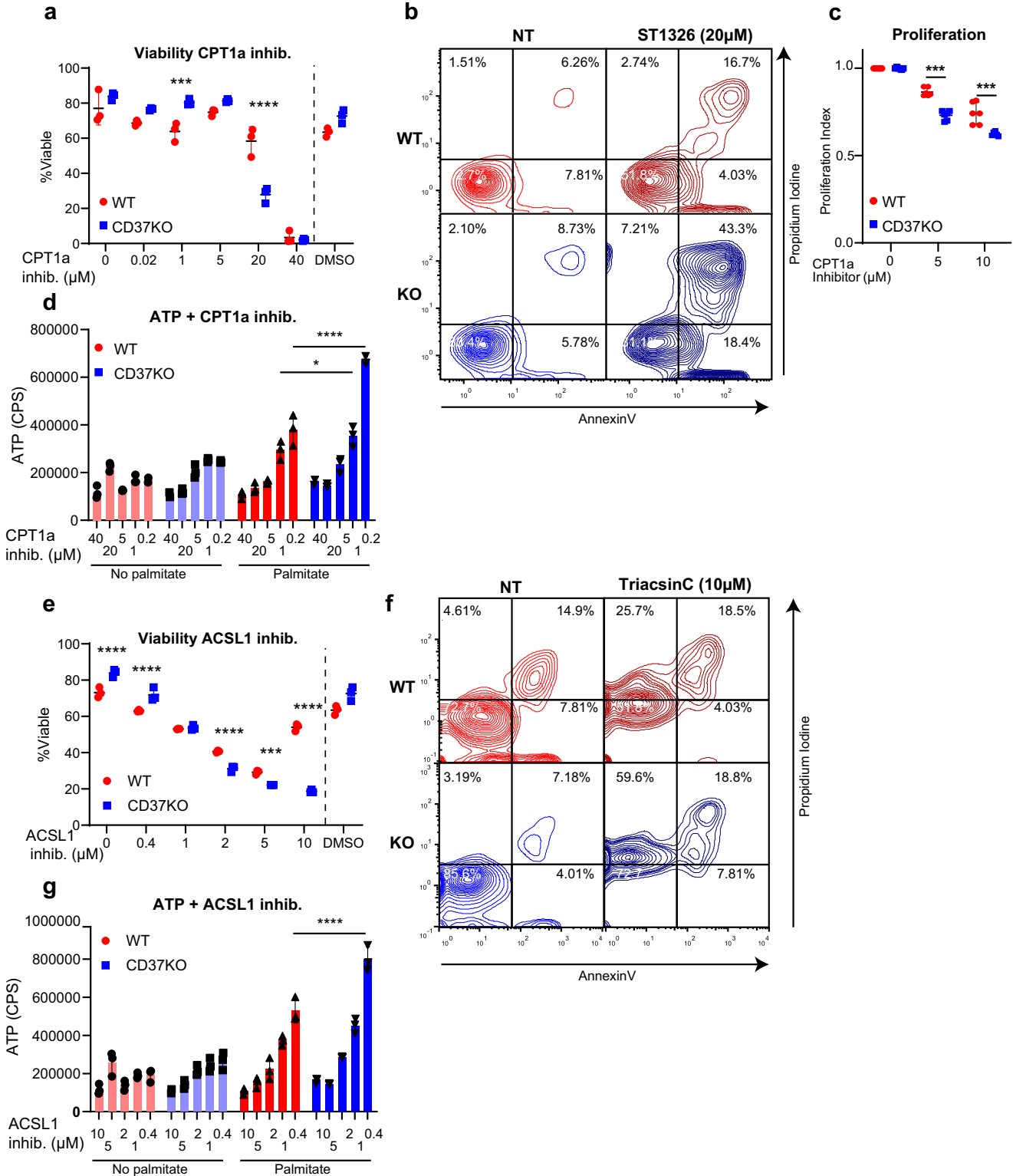

**Fig. 6 | CD37KO B-cell lymphomas are dependent on the fatty acid metabolic switch and can be therapeutically targeted using FA inhibitors.** Effects of 24 h CPT1a inhibition with ST1326 at indicated concentrations on viability ($n = 3$, 1 μM; $p = 0.0031$, 20 μM; $p < 0.0001$), apoptosis, necrosis (**a**, **b**) proliferation capacity ($n = 6$, 5 μM; $p = 0.0003$, 10 μM; $p = 0.0009$) (**c**) and on palmitate-dependent energy production by WT and CD37KO lymphoma cells ($n = 3$, 1 μM; 0.0297, 0.2 μM; $p < 0.0001$) (in counts per seconds, CPS) (**d**). Effects of 24 h ACSL1 inhibition with TriacsinC at indicated concentrations on viability (0, 0.4, 2, and 10 μM; $p < 0.0001$,

5 μM; $p = 0.0005$), apoptosis, necrosis (**e**, **f**) and on palmitate-dependent energy production by WT and CD37KO lymphoma cells ($n = 3$, 0.4 μM; $p < 0.0001$) (in counts per seconds, CPS) (**g**). Two-way ANOVA with Tukey's post hoc test revealed significant differences between the indicated groups, *$p < 0.05$, **$p < 0.01$, ***$p < 0.001$ ****$p < 0.0001$. Error bars represent mean ± SD. Experiments were repeated twice, yielding similar results. Source data are provided as a Source Data file. See also Supplementary Fig. 6.

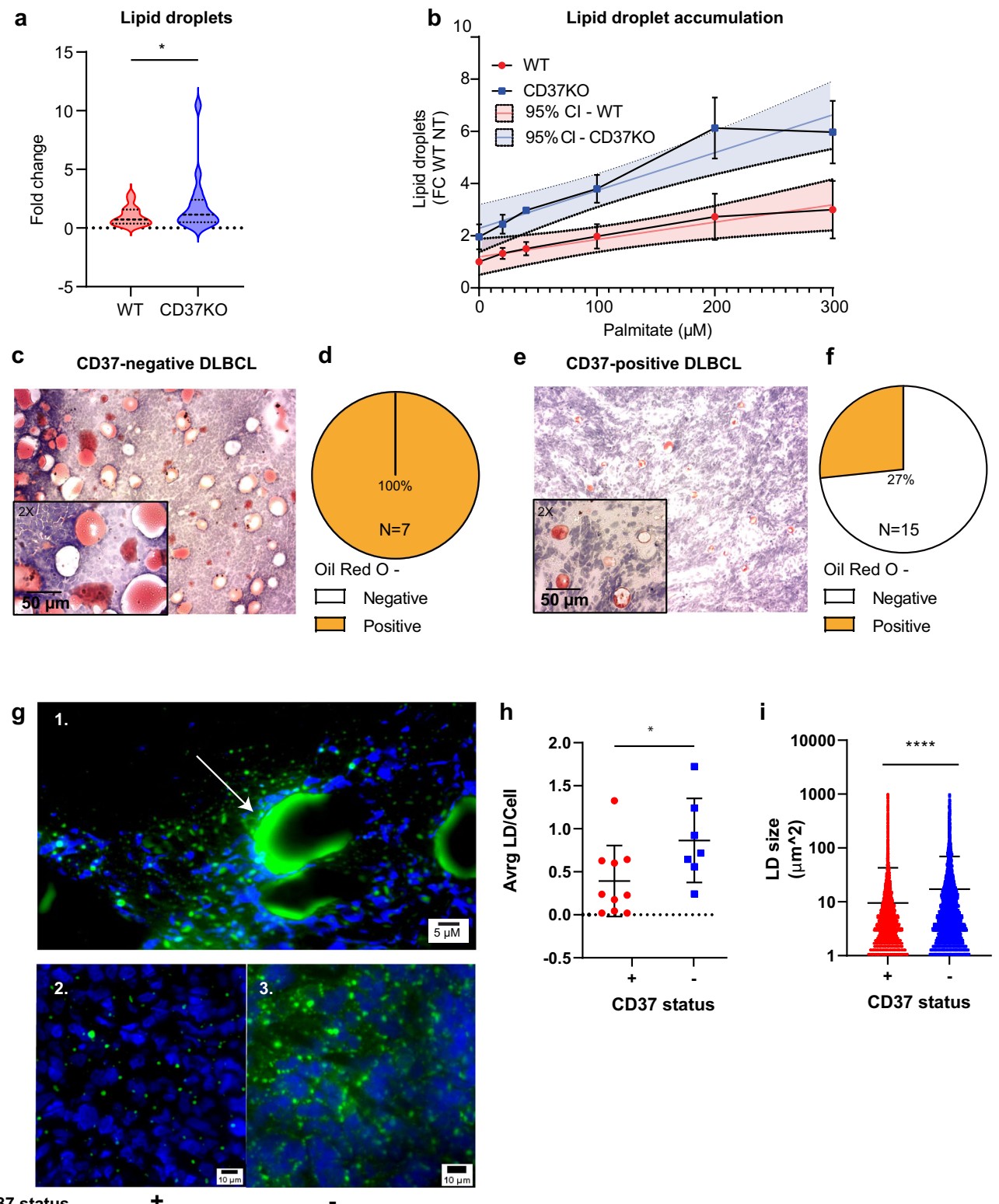

forming potential of cancers has been associated with increased survival in hypoxic environments such as the light zone of germinal centres[3,82,83]. Both lipid droplet formation and TCA maintenance are drastically enhanced in CD37-deficient lymphoma and premalignant B cells, which may be of particular benefit within the germinal centre environment. FATP1-knockout mice show smaller lipid droplets in adipose tissue and fail to retain their core body temperature, despite elevated serum free

fatty acid levels, confirming the importance of FATP1 for lipid droplet formation[84]. Furthermore, knockdown of CPT2 resulted in a diminished germinal centre B-cell pool, further confirming an advantage for FA catabolizing, healthy B cells to proliferate in germinal centres[37,85]. Interestingly, in inherited disorders of mitochondrial fatty acid oxidation, increased lipid storage was observed as a result of impaired fatty acid degradation e.g. in the liver and heart[86]. Here, the formation of lipid droplets may also be

**Fig. 7 | CD37-deficient lymphoma cells and primary human lymphomas have enhanced lipid storage capacity.** Human WT and CD37KO lymphoma cells (BJAB) were stained with Bodipy 493/503 that accumulates in neutral lipid regions found in lipid droplets. Quantification of LDs in untreated conditions ($n = 16$, $p = 0.0257$) (**a**) and palmitate supplemented (**b**) human WT ($n = 4$) and CD37KO ($n = 2$) lymphoma cells using flow cytometry. Experiments were repeated three times, yielding similar results. Tissues of primary CD37-negative DLBCL (**c**) and CD37-positive DLBCL (**e**) were stained with oil red O (ORO) and counterstained with haematoxylin. Quantification of ORO-positive tumour tissues from CD37-negative ($n = 7$) (**d**) and CD37-positive ($n = 15$) (**f**) DLBCLs. Same tissues were also stained with Bodipy 493/503 for 30 min and visualised using fluorescent microscopy to show the proximity of cells with LDs to large lipid deposit (1. arrow) and a zoom of CD37 positive (2.) and negative (3.) cells (**g**). Three large zones were selected from each total-section tile scan and analysed for the size (CD37pos: $n = 27357$ LD in ten individual tissue sections, CD37neg: 6827 LD in seven individual tissue sections, $p = 0.0482$) (**h**) and amount relative to nuclei (**i**) of the lipid droplets. One-way paired t-test (**a**) and two-way unpaired t-test (**h**, **i**) revealed significant differences between indicated groups, $*p < 0.05$, $****p < 0.0001$. Blue (squares) and red (circles) zones (**b**) represent the 95% Confidence interval for the trend. Error bars represent mean ± SD. Source data are provided as a Source Data file. See also Supplementary Fig. 7.

linked to cytoprotective effects by enabling coordinated oxidation and storage of fatty acids[87]. Thus, the finding of lipid droplet build-up concomitant with stimulated FAO in CD37-deficient cells may represent an adaptive advantage.

Finally, the enhanced lipid droplet formation in CD37-negative DLBCL tissues of patients supports the translational value of our present findings. Studies with therapeutical interventions in fatty acid metabolic pathways, such as CPT1a inhibition in (haematological) cancers are ongoing and show promising results[68,88,89]. Taken together, our study contributes key insights to the fast-growing, yet understudied, field of cancer metabolism and identifies CD37 as a gatekeeper of a fatty acid metabolic switch in aggressive B-cell lymphoma.

## Methods

All research complies with all relevant ethical regulations of the Radboud Institute for Molecular Life Sciences. All murine studies complied with European legislation (directive 2010/63/EU of the European Commission) and were approved by local authorities (CCD, The Hague, the Netherlands) for the care and use of animals with related codes of practice.

### Mouse studies and B-cell isolations

*Cd37*-deficient (CD37KO) mice and wild-type (WT) littermates on a C57BL/6 J background matched for age and gender were used as described[20]. Mice were kept in enriched shelters in a pathogen-free facility with 12 h:12 h light:dark cycle with temperatures between 18–22 °C and a humidity between 40–60%, and had ad libitum access to food (R/M H chow, Sniff, Soest, Germany) and water. Mice were checked daily for general condition and human endpoints included a drop in general condition or cachexia, or loss of weight of more than 15% in 2 days. Mice were euthanized using cervical dislocation at age 4-6 months old. Splenectomy and subsequent CD43-negative MACS isolation were performed under sterile conditions, and splenocytes were randomly assigned to experimental groups. Single-cell suspensions were made from the spleen and lymphocytes were isolated via haemolysis and subsequent density separation with Lympholyte-M (Cedarlane Laboratories, Burlington, Canada). B cells were isolated via negative selection with magnetic CD43-coated beads (Miltenyi Biotec, Bergisch Gladbach, Germany). B220 antibody (5 μg/mL, RA3-6B2, conjugated) (BioLegend, San Diego, CA, USA) staining was used to assess purity using flow cytometry (MACS Quant, Miltenyi Biotec, Bergisch Gladbach, Germany). Only isolations with 90–100% purity were used in this study.

### Lipidomics studies of murine serum and premalignant B cells

Serum was collected from CD37KO mice and WT littermates (aged 16–24 months old, with or without lymphoma) for lipidomic analysis (six healthy mice/genotype and three tumour-bearing mice/genotype). Cell pellets were isolated from wild-type and CD37KO spleens as described above. To dried cell pellets ($5e^6$ cells/mouse, normalised to cell count) and serum (25 μL) of WT and CD37KO mice, the following was added: 100 μL water, 160 μL methanol, 25 μL internal standard mix (Sciex, Lipidyzer kit) and 575 μL methyl-tert-butylether (MTBE). All samples were vortexed for 5 s, sonicated for 1 min and then vortexed

again for 5 s. To complete lipid extraction, all samples were left at room temperature for 30 min. Next, all samples were centrifuged at $18.213 \times g$ for 5 min at 20 °C. From each sample, 750 μL of supernatant was taken and transferred to a 2 mL Eppendorf tube. Extractions were repeated by adding 300 μL of MTBE and 100 μL of MeOH. All samples were vortexed and centrifuged at $18,213 \times g$ for 15 min at 20 °C. The organic extracts were combined and 300 μL of water was added for phase separation with subsequent centrifugation at $18,213 \times g$ for 5 min at 20 °C. From the upper organic layer, 650 μL of supernatant was transferred to a 1.5 mL glass vial. The organic extract was dried at room temperature under a gentle stream of nitrogen and 250 μL running buffer (50:50 methanol:dichloromethane, MeOH:DCM with 10 mM ammonium acetate) were added to the dried samples. Finally, the samples were to a glass vial for injection. For serum, the identical procedure was applied, using 25 μL of the sample. Further lipidomics analysis was performed as described previously[34]. As plasma has a density of 1.025 g/mL these data are in practice 1:1 translated into nmol/mL. A detailed description of all calculations can be found here[90], and on the following Github page https://github.com/syjgino/SLA. Datafiles are accessible to everyone in the Source Data.

### Carnitine flux studies

To determine intracellular acylcarnitines, primary WT and CD37KO splenocytes were isolated as described above, washed with PBS and stored at −80 °C until further processing. Cell lysates (0.5-1e⁶ cells) were prepared by resuspending the cells with 150 μL of buffer (PBS supplemented with 1% of protease inhibitor cocktail, P8340, Merck, Darmstadt, Germany), subjecting the cell suspension to three cycles of freeze-thawing in dry-ice/room temperature followed by three cycles of sonication (pulse duration, pause, on ice, instrument used). About 50 μL of cell lysate was then processed by methanol extraction according to standard operating practices for the determination of acylcarnitine profiles for diagnostic purposes in the Laboratory of Clinical Biochemistry and Metabolism, Centre of Metabolism Freiburg. Acylcarnitines were determined via flow-injection into an ultra-performance liquid chromatography system (Acquity UPLC, waters, Massachusetts, USA) coupled to a tandem mass spectrometer (MS/MS, Quattro Premier XE, Waters, Massachusetts, USA). Comparison of acylcarnitine profiles was simplified by pooling the species according to their acyl group chain length, as follows: short chain (C-C5), medium chain (C6-C12) and long chain (C12-C18). All metabolite concentrations were normalised by the total concentration of protein in the cell lysate, determined using the BCA assay (Pierce, Thermo Fisher Scientific, Waltham, MA, USA).

### Quantification of fatty acid transporter and FA-metabolism associated gene expression in primary murine B cells

RNA from three WT and three CD37KO murine CD43−, B220+ B cells was isolated with an RNeasy kit (Qiagen, Hilden, Germany). Samples were enzymatically fragmented and biotinylated using the WT Terminal Labelling Kit (Affymetrix Thermo Fisher Scientific, Waltham, MA, USA). ChIP (GPL6246) was scanned with the Affymetrix Gene ChIP Scanner 3000 7G to conform manufacturer's instructions (Affymetrix Thermo Fisher Scientific, Waltham, MA, USA). The data discussed in

this publication have been deposited in NCBI's Gene Expression Omnibus[91] and are accessible through GEO Series accession number GSE212032 (https://www.ncbi.nlm.nih.gov/geo/query/acc.cgi?acc=GSE212032).

### Studies with human cells and tissues

**Human cell line handling.** Human lymphoma BJAB (Cat: ACC 757), OciLy1 (Cat: ACC 722), OciLy8, WSU-NHL (Cat: ACC 58) (CD37-positive) and OciLy19 (Cat: ACC 528), SUDHL6 (Cat: ATCC CRL-2959) (CD37-negative) cells were cultured in RPMI1640 medium supplemented with 10% heat-inactivated foetal bovine serum (FBS), 1% antibiotic-antimycotic (AA), and 1% ultra-glutamine (UG) (Gibco, Thermo Fisher Scientific, Waltham, MA, USA) and maintained at 37 °C with 5% $CO_2$. Cells were seeded at ~$3e^5$ cells/mL on the day prior to experiments. CD37-knockout (CD37KO) BJAB and OciLy1 cells were generated by CRISPR/Cas9 technology as described[92]. Cell lines were routinely checked for the absence of mycoplasma contaminations. Cell lines were derived from DSMZ, ATCC and Dr. Blanca Scheijen (Dept. Pathology, Radboudumc) and cell lines were authenticated using STR-analysis.

**Seahorse metabolic analyser studies.** Metabolic respiratory assays were performed as described by the Seahorse manufacturer (Agilent Technologies, Santa Clara, CA, USA). Cells were seeded on Cell-Tak (Corning Incorporated, New York, NY, USA) coated Seahorse cell culture plates and exposed to Seahorse XF Base Medium supplemented with 2 mM glutamine (For measuring extracellular acidification rate, ECAR), or 1 mM pyruvate, 2 mM glutamine, and 10 mM glucose (For measuring oxygen consumption rate, OCR). Cells were subsequently rested in 0% $CO_2$ at 37 °C for 45–60 min before measuring.

**Energy quantification studies.** ATP luminescent reporter assays were performed in full growth medium (10% FBS, 1% UG, 1% AA) and supplemented with indicated substrates (10 mM glucose and 10−20 μM palmitate) and inhibitors (as indicated) for 1–2 h (or more if indicated) at 37 °C. Lymphoma cells were lysed with 5% trichloric acid (TCA) (Thermo Fisher Scientific, Waltham, MA, USA) and spun down at 10,000×$g$ for 15 min. Supernatants were buffered with TRIS (pH 7.4) (Thermo Fisher Scientific, Waltham, MA, USA). About 50 μL was transferred and measured in triplicate by Cellstar Bioluminescence measure (Gibco, Thermo Fisher Scientific, Waltham, MA, USA).

**Substrate uptake studies.** Uptake of glucose and palmitate were assessed with fluorescent analogues 2-(N-(7-Nitrobenz-2-oxa-1,3-diazol-4-yl)Amino)-2-Deoxyglucose (2-NBDG, Thermo Fisher Scientific, Waltham, MA, USA, N13195), Bodipy FL C12 (D3822) and Bodipy FL C16 (D3821) (Thermo Fisher Scientific, Waltham, MA, USA) using flow cytometry (MACS Quant, Miltenyi Biotec, Bergisch Gladbach, Germany). Here, 20 nM 2-NBDG and 40 nM Bodipy FL C12 or C16 (in PBS) was added to $1e^6$ lymphoma cells or 95% pure CD43-, B220 + primary B cells for 15 min at 5% $CO_2$ at 37 °C. eFluor 450 (1:2000, Thermo Fisher, Waltham, MA, USA, 65-0863-14) viability dye was used to determine viable cells. Cells were washed before measurement by FACS (MACS Quant, Miltenyi Biotec, Bergisch Gladbach, Germany).

**Fatty acid metabolism inhibition studies.** Inhibition assays were performed in a range of 0.125 to 12.5 μM FATP1 inhibitor, compound 5k and 12a[43,44] (kindly provided by Tsuyoshi Shinozuka, Japan), 0.02 to 40 μM CPT1a inhibitor, ST1326 (Avanti Lipids, Croda International, Snaith, UK), 5 μM CPT1a inhibitor, Etomoxir (Cayman Chemical, Michigan, USA) and 0.4 to 10 μM ACSL1 inhibitor, Triacsin C (Abcam, Cambridge, UK) in RPMI1640 with 10% FBS, 1% UG, and 1% AA. For the compound 5k studies, WT and CD37KO cells of the same genotypic background (BJAB and OciLy1) were pooled prior to treatment, and later identified by CD37 FACS-signal to exclude differences in culture conditions. About 5 μM CellTrace CFSE (Thermo Fisher Scientific, Waltham, MA, USA) was used to assess proliferation. Cell division numbers were established by assessing peak intensities. AnnexinV/7-AAD (1:100, BioLegend, San Diego, CA, USA, 640930) stainings were used to determine viability after inhibition using flow cytometry (MACS Quant, Miltenyi Biotec, Bergisch Gladbach, Germany).

**Transporter quantification and colocalization studies.** Antibody stainings for flow cytometry (MACS Quant, Miltenyi Biotec, Bergisch Gladbach, Germany) and confocal microscopy (LSM900, Zeiss, Jena, Germany) were performed on cells fixed with 2% paraformaldehyde and blocked with 5 μM FcBlock (Bd Biosciences, San Jose, CA, USA). Antibodies directed against FATP1 (308420, conjugated) (R&D Systems, Minneapolis, MN, USA, IC3304R), CD36 (877302, conjugated) (Novus Biologicals, Centennial, CO, USA, MAB19553AF647) and CD37 (HH1, unconjugated) (Santa Cruz Biotechnology, Dallas, TX, USA, sc-18881) were added as 10 μg/mL in PBS and stained for 30 min at RT in the dark. Isotype controls were added in similar concentrations. A secondary antibody against CD37 was used as 2,5 μg/mL in PBS and stained for 30 min at RT in the dark. For colocalization studies (LSM900, Zeiss, Jena, Germany), the Coloc2 function in ImageJ was used (Version 1.53 g)[93]. Proximity ligation assays (PLA) were performed with FATP1 and CD37 antibodies according to the manufacturer's instructions (Merck, Darmstadt, Germany). In addition, a second FATP1 antibody (polyclonal) (Novus Biologicals, Centennial, CO, USA, NBP2-69016) was used at 10 μg/mL as a positive control. CD37KO BJAB cells (generated by CRISPR/Cas9 knockout, described in[92]) were used as a negative control in these experiments. Analysis was performed with ImageJ[42,94].

**Rescue experiments CD37.** CD37-WT-psGFP2-N1 (2 μg/mL) constructs were transfected into CD37KO BJAB lymphoma cells using the Neon™ transfection system (Thermo Fisher Scientific, Waltham, MA, USA) according to the manufacturer's instructions. Cells were resuspended in RPMI1640 with 10% heat-inactivated foetal bovine serum (FBS) and 1% ultra-glutamine (UG) (Gibco, Thermo Fisher Scientific, Waltham, MA, USA) and maintained at 37 °C with 5% $CO_2$ for 24 h after transfection.

**$^{13}$C-Palmitate-BSA conjugation.** A 10% FFA-free BSA (Merck, Darmstadt, Germany) solution was made and filter sterilised. Subsequently, a 20 mM palmitate solution was made by carefully dripping FA stock (200 mM) (605573, Merck, Darmstadt, Germany) solution in pre-warmed water without touching the pipet tip to the water. Sterile filtered KOH 1 M was added while shaking the solution slowly until it became clear. The 10% BSA solution was then added to the FA solution to achieve a 10 mM end concentration. The FA was conjugated to BSA for 30–45 min at 37 °C and mixed carefully every 10 min.

**Carbon tracing studies.** $^{13}$C tracer experiments were performed as described before (Zaal et al., 2017). Human lymphoma WT and CD37KO BJAB cells were cultured in RPMI1640 containing 20 μM [U-$^{13}$C]D-Palmitate at a density of $3e^5$/mL. After 4 and 8 h, samples were washed with PBS and harvested by centrifugation for 5 min at 1000 × $g$ at 4 °C. Metabolites were extracted by adding 100 μL ice-cold MS lysis buffer (methanol/acetonitrile/uLCMS $H_2O$ (2:2:1)) to the cell pellets. Samples were shaken for 10 min at 4 °C, centrifuged at 14,000 × $g$ for 15 min at 4 °C and supernatants were collected for liquid chromatography-mass spectrometry (LC-MS) analysis. LC-MS analysis was performed on a Q-Exactive HF mass spectrometer (Thermo Fisher Scientific, Waltham, MA, USA) coupled to a Vanquish autosampler and pump (Thermo Fisher Scientific, Waltham, MA, USA). Metabolites were separated on a Sequant ZIC-pHILIC column (2.1 × 150 mm, 5 μm, Merck, Darmstadt, Germany) with a guard column (2.1 × 20 mm, 5 μm, Merck, Darmstadt, Germany) using a linear

gradient of acetonitrile and 20 mM (NH4)2CO$_3$, 0.1% NH$_4$OH in ULC/MS grade water with a flow rate of 100 μL/min. The MS operated in polarity-switching mode with spray voltages of 4.5 and −3.5 kV. Metabolites were identified based on exact mass within 5 parts per million (ppm) and further validated by concordance with retention times of external standards. Metabolites were quantified using Tracefinder software (Thermo Fisher Scientific, Waltham, MA, USA). Peak intensities were normalised based on median peak intensity. Isotopomer distributions were corrected for natural [13]C abundance in samples supplemented with [12]C-Palmitate.

**Co-IP and Western-Blot for CD37 and FATP1.** For co-immunoprecipitation of FATP1 and CD37, BJAB CD37KO cells were transfected with constructs for FLAG-FATP1 and alpha-tag-CD37 both in pCDNA3.1 using the Amaxa 4D nucleofection system (SF Cell Line 4D-NucleofectorTM X Kit L, programme CV-104). The next day, cells were collected and washed with PBS and solubilized in a buffer containing Brij 97 (1% Brij 97, 25 mM Tris-HCl pH 7.5, 150 mM NaCl, 2 mM EDTA and phosphatase and protease inhibitors) for 30 min with regular agitation at 4 °C. Non-solubilized material was removed by centrifugation for 10 min at 4500×*g*. The supernatant was precleared by incubating with Sepharose CL-4B beads (G-Biosciences, St. Louis, MO, USA) for 30 min while rotating, followed by centrifugation at 10,000 × *g* for 10 min. Next, the precleared solution was incubated with Alpha Selector beads (Nano Tag Biotechnologies, Göttingen, Germany) for 2 h while rotating to pulldown alpha-tagged CD37. The beads were washed extensively with wash buffer containing 0.1% Brij 97, 25 mM Tris-HCl pH 7.5, 150 mM NaCl, 2 mM EDTA and phosphatase and protease inhibitors. Proteins were eluted with SDS sample buffer and analysed by separation on a 10% SDS-PAA gel followed by transfer to PVDF. The blot was blocked with Intercept-TBS (Li-Cor, Lincoln, USA) for 1 h at room temperature and incubated with mouse anti-FLAG antibody (M2, Sigma/Merck, Darmstadt, Germany, F3165) at a dilution of 1 μg/ml. FLAG-FATP1 was detected with goat anti-mouse IRDye 800 (Li-Cor) and the blot was scanned with a Typhoon laser scanner (Amersham). Immunoprecipitated alpha-tagged CD37 was visualised by stripping the blot and reprobing with rabbit anti-CD37 (GeneTex, Irvine, CA, USA, GTX129598), followed by incubation with donkey anti-rabbit IRDye 800 and imaging with Typhoon laser scanner. N-terminal tagged human FATP1 was obtained from Genscript. N-terminal tagged CD37 was obtained by inserting the alpha tag in the ORF of human CD37 using the Q5 Site-Directed Mutagenesis Kit (NEB). Sequences were verified by Sanger sequencing.

**Lipid staining lymphoma cell lines and primary tumour tissues of diffuse large B-cell lymphoma patients.** WT and CD37KO cells were washed with PBS and resuspended in 2 μM Bodipy 493/503 (Thermo Fisher Scientific, Waltham, MA, USA) in PBS with 0.5% foetal cow serum. Cell suspensions were kept at 37 °C and 5% CO$_2$ for 15 min. Suspensions were then washed and analysed by flow cytometry (MACS Quant, Miltenyi Biotec, Bergisch Gladbach, Germany). Frozen tissues of diffuse large B-cell lymphoma patients were sectioned at 4 μm and air-dried for 30 min prior to staining. Sections were submerged in prediluted Oil Red O in 0.5% propylene glycol (Merck, Darmstadt, Germany) for 60 min at RT, or in 2 μM Bodipy 493/503 for 30 min. Sections were then briefly and carefully rinsed with demineralised water and subsequently counterstained with Haematoxylin (Merck. Darmstadt, Germany) or nuclear dye DAPI (for ORO and Bodipy samples, respectively) for 10 min. After counterstaining, the sections were rinsed with a continuous water stream. Finally, sections were embedded and enclosed with pre-heated glycine-gelatine.

**Statistics and reproducibility.** All source data is provided within this paper. Statistical analysis was performed using GraphPad Prism 8 software. Each statistical test performed is indicated in the associated figure legends. Experiments with human tumour material were performed blinded. No statistical method was used to predetermine sample sizes. Data obtained using the Seahorse metabolic analyser were checked for proper injection of the inhibitors or uncouplers, and wells that did not show response to inhibitors or uncouplers were excluded from analysis. In all figure legends, we identify the meaning of the error bars, the amount of biologically independent groups, the exact *p* values and the number of independent experiments.

### Reporting summary

Further information on research design is available in the Nature Research Reporting Summary linked to this article.

## Data availability

The lipidomics data generated in this study have been made available in the Source Data and are accessible to everyone. The RNA-data discussed in this publication have been deposited in NCBI's Gene Expression Omnibus[91] and are accessible through GEO Series accession number GSE212032. All raw data are available in the main article, supplementary information, and source data file. Source data are provided with this paper.

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

## Acknowledgements

A.B.v.S. is supported by the Netherlands Organization for Scientific Research NWO Gravitation Programme 2013 grant (ICI 00023), ZonMW (project 09120012010023), the Dutch Cancer Society (KWF) (11618/2018 and 12949/2020), and was awarded a European Research Council Consolidator Grant (Secret Surface, 724281). C.R.B. is supported by grant ICI-00014 from the Institute for Chemical Immunology, funded by the NWO Gravitation Programme. R.S. was supported by an IMI grant (Hyporesolve) and a grant from Health Holland (Timid). N.B. and M.G. were partially supported by NWO project 184.034.019. The authors thank M. van den Brand for providing the patient DLBCL tissue samples, I. van Raaij for the Oil red O staining protocol and control samples. The authors are very grateful to Tsuyoshi Shinozuka for providing compound 5k and 12a.

## Author contributions

A.B.v.S. and R.P. conceived the presented idea with additional input from R.S., L.H., S.J.v.d., J.J., M.t.B. and C.R.B.; R.P. performed the Seahorse metabolic assays, ATP quantifications, FACS-experiments and microscopy with additional help from J.C.E., A.C.G.B. and M.V.M.; M.G. and N.B. performed the lipidomics; L.H and U.S. performed carnitine-quantification; E.A.Z., A.T.H. and C.R.B. performed carbon tracing studies; M.t.B. performed Co-IP; R.P. did the data analysis and wrote the manuscript with supervision from A.B.v.S. and critical feedback from all authors.

## Competing interests

There are no relevant financial or non-financial competing interests.
