## [Peer Review File · Nature Communications]

Fatty acid metabolism in aggressive B-cell lymphoma is inhibited by tetraspanin CD37REVIEWER COMMENTS

Reviewer #1 (Remarks to the Author):

In this manuscript, Peeters et al. find that CD37, a tetraspanin protein family member associated with poor prognosis in aggressive lymphomas, is a negative regulator of fatty acid (FA) metabolism. Using a combination of lymphoma cell line and mouse knockout models, they show that CD37 deficiency is associated with enhanced palmitate uptake and metabolism, as well as increased lipid droplet (LD) formation. The effect of CD37 on FA metabolism are proposed to be mediated by the inhibitory interaction between CD37 and the FA transporter FATP1 at the plasma membrane. Overall, the studies provide new information on the role of CD37 in lymphomagenesis through regulation of FA metabolism. They also identify FATP1 as a novel molecular effector of CD37 beyond previously identified transporters and signaling pathways. A significant limitation of the current manuscript is the lack of evidence that FATP1 is functionally required for the effect of CD37 deficiency on FA metabolism. In addition, the studies implicating altered LD formation in the context of CD37 deficiency are mostly correlative and underdeveloped.

Specific comments:

- 1) The current studies lack critical test of FATP1 as the functional link between CD37 and regulation of FA metabolism. The PLA data in support of complex formation between CD37 and FATP1 are intriguing and the downstream manipulation of CPT1 and ACSL1 show FAO is relevant for proliferation and survival of CD37 KO cells but do not directly point to the functional requirement for FATP1 in this setting. The author should test the consequences of FATP1 inhibition/knockdown/knockout on FA metabolism in the context of CD37 negativity. This is important given the many downstream targets of CD37 beyond FATP1 that can regulate FA metabolism. For example, the authors have previously implicated IL-6 signaling in CD37 negative lymphomas. IL-6 can enhance lipolysis and FAO.
- 2) The studies related to LDs in Fig. 7 are underdeveloped, and do not coherently flow with the rest of the studies in the paper. In their current version, they predominantly provide correlative data. CD37 deficient cells appear to respond to palmitate both via FAO (Fig. 1 and 4) and LD formation (Fig 7b). The idea of LDs as an adaptive mechanism against lipotoxicity is interesting and has been shown in other cell types as well. However, the functional relevance of these observations/ideas in CD37-dependent regulation of substrate switching is unclear and has not been adequately addressed. How relevant is lipotoxicity in the biology of germinal center B cells/lymphomas? Alternatively, are LDs predominantly a reservoir of fuel for these cells? The authors should either provide a more coherent integration of these observations with the rest of the paper or consider removing these data from the current manuscript.
- 3) The proposed mechanism of CD37's effect on FA metabolism suggests specificity for palmitate within LCFA species. The biochemical basis for this specificity will strengthen the manuscript. Is this specificity strictly dictated by FATP1? This is relevant given the lack of consensus in the field surrounding the exact role of FATP1 in FA metabolism. In addition, the authors should strengthen/validate the selectivity of the proposed mechanism for palmitate by using another FA as a negative control for palmitate in feeding the TCA cycle metabolites (Fig. 4) and FA uptake studies (Fig. 5).
- 4) The authors selected BJAB and OCyLy1 as human cell line models for CD37 knockout, Seahorse and metabolic tracing studies (although only BJAB data are shown/discussed). It will be helpful to include the rationale behind selecting these particular cell lines in terms of their starting CD37 status. As the authors know, certain human DLBCL cell lines readily oxidize palmitate (Pfeiffer, K422, etc). Are these lines negative for CD37? Is gain-of-function in CD37 sufficient to block FAO in these DLBCL cell lines?
- 5) Related to the above point, what is the FAO capacity of the CD37 negative primary DLBCLs in Figure 7 compared to CD37 positive counterparts?

Additional points:

- 6) Fig. 6 panels g, h and j- CPT1 and ACSL1 inhibition also have substantial effect on viability, proliferation and necrosis of WT cells. The authors should include stats for these comparisons as well.
- 7) A rationale for why the authors specifically measured necrotic death (as opposed to other forms of cell death) will improve the flow of the text. It is also unclear why the authors chose different

readouts for CPT1 vs ACSL1 inhibition (survival and proliferation for one, ATP and necrosis for the other). The description of these results (lines 250-251) is also confusing as the effect of CPT1 and ACSL1 inhibition on necrosis is compared in the text but Fig. 6j only involves ACSL1 inhibition.

8) It appears that in all seahorse experiments, the media contains 2 mM glutamine at minimum. As such, referring to this as nutrient deprived condition does not seem accurate.

Reviewer #2 (Remarks to the Author):

In this study, the authors provide evidence that CD37-negative B cell lymphomas undergo a metabolic switch from glycolysis to fatty acid oxidation, in particular by using palmitate as an energy source. Supporting their hypothesis, they provide extensive metabolomic data on CD37-negative and control B cell lymphomas, and data supporting an interaction between CD37 and the fatty acid cell surface transporter, FATP1. The potential impact of the study is high because it would lay the foundation for understanding the aggressive behavior of CD37-negative B cell lymphomas and point to potential therapeutic interventions targeting fatty acid utilization as an energy source. There are a few major issues that need to be addressed to make the case for the central working hypothesis advanced by the paper.

1. Evidence for a direct CD37-FATP1 interaction is based only on proximity ligation assay (PLA). It is unclear that PLA has the spatial resolution to distinguish direct from indirect protein-protein interactions, but the authors repeatedly claim it as evidence for a direct interaction.
2. The PLA assay needs additional controls. While the fact that no PLA signal is seen between CD37 and FATP1 antibodies in CD37-negative cells is reassuring regarding non-specific binding of the CD37 antibody, a stronger negative control is required. For example, showing that a GPI-linked protein such as CD55 does not generate a PLA signal with CD37 or FATP1, would increase confidence in the specificity of the assay. Alternatively, demonstrating the CD37-FATP1 interaction by a second additional method would increase confidence in the interaction.
3. The authors claim that CD37 negatively regulates FATP1 activity. This is an appealing hypothesis, but no direct evidence is provided to support it. Uptake of palmitate analog Bodipy FL C16 is reduced in CD37-positive cells, but the proportion of the uptake mediated by FATP1 was not measured. Moreover, to make a strong claim that CD37 is functioning by directly regulating FATP1, something like expressing a FATP1 mutant incapable of CD37 interaction or some other type of structure-function experiment would be required.

Some additional issues include

4. In some statistical analyses, WT control data are flattened to equal exactly 1.0, but then used for statistical testing (Fig. 3f,g,h, Fig. 5a and Fig. 7a). The t-test is influenced by the spread in the data, but this has been artificially removed for the WT control data in these panels. The WT control data points could each be divided by WT control overall mean to restore the variability in the WT control data points to what was actually observed in the experiments, and then the t-test could be used.
5. In Fig. 5F, how is it possible to have a negative number of counts per second in luminescence assay for ATP?
6. In the IHC data in Fig. 7c&d, it's not clear how the large fatty deposits in the CD37-negative tissue specimen (many times larger than individual cells?) relate to lipid droplets that accumulate inside cells.

Reviewer #3 (Remarks to the Author):

This manuscript by Peeters et al. examines metabolic alterations in nutrient utilization in B cell lymphoma induced by loss of the tetraspanin, CD37. This group has previously shown that loss of CD37 expression in this cancer leads to a poorer prognosis and this makes the clinical relevance of

this manuscript high. However, some of the major metabolic findings of the manuscript show high variability and refinement of data presentation is needed.

Major comments:

1. The authors perform lipidomics (Fig 2) on CD37 KO mice and WT mice with and without B cell lymphoma. Examining the serum, they don't see overall changes in serum lipid levels but examination of individual species shows changes in TAGS and FFA. However, when FFA were quantified only palmitate shows a statistically significant difference. From the heat maps shown in figure 2 there appears to be significant variability between biological replicates especially WT CD37KO without tumor. The lack of correlation between the lipid quantification and the fold changes shown in Figure 2d are confusing. FFA and lipids in serum are heavily dependent on diet and no methods are described for controlling whether the mice were fed or fasted prior to the lipidomics analysis and whether the large tumor burden in the CD37KO impacted their ability to eat (do they exhibit weight loss over the course of disease). It seems unlikely that there would be preferential uptake of palmitate for beta oxidation versus other long chain fatty acids as they are equally likely to enter into this pathway. There are also significant differences in other LCFA just examining CD37KO animals to WT. Could the authors resolve this discrepancy with additional biological replicates and control of feeding?

2. Similarly, in the isolated B cells there is the same mismatch between the heat maps and lipid quantification. This again may be due to variability between biological replicates, particularly since one CD37KO sample seems to show larger increases in C16, C20, C20:4, and C18 compared to other replicates and this may skew the results. Since the authors show that CD37 interacts with FATP1 which is a general LCFA transporter it seems unlikely that there is specificity towards palmitate. The overall changes in total FFA shown in 3C likely reflect changes in the levels of other lipids that aren't captured in the statistical analysis of quantification of individual species due to variability.

3. The authors claim that the changes in short chain and medium chain acylcarnitines reflects increased beta oxidation may not be correct. Additionally, the schematic in 4B is incorrect. Short chain and medium fatty acids do not require conjugation to carnitine to be transported into the mitochondria, rather they diffuse through the membrane. Carnitine conjugation is used for transport into the mitochondria and once in the mitochondria acylCoA species are formed via CPT2. The acylCoA species then undergo beta oxidation. It is incorrect to show that the acylcarnitine species undergo beta oxidation in figure 4B. Are these excess SCFA and MCFA carnitine species just a byproduct of excess lipids in CD37KO cells?

4. The isotope tracing experiments shown in Figure 4 and Supplemental Figure 3 are confusing and the data presentation is not clear. The authors should show full isotopologue patterns for TCA metabolites and palmitoylcarnitine as these are the primary metabolites that should be labeled when cells are fed labeled carnitine. Additionally, showing fold changes in heat maps instead of labeling patterns is confusing and not typically how isotope labeling experiments are presented. Are the authors suggesting that labeled butyryl carnitine arises from incomplete oxidation of palmitate in the mitochondria and subsequent conjugation to carnitine? The transport of acylcarnitines into mitochondria is driven by favorable free energy changes due to beta oxidation; ensuring transport of FA into mitochondria and complete oxidation. If the authors are proposing that labeling of SCFA-carnitines are occurring due to incomplete oxidation they need to show that.

5. The functional analyses with the CPT1a inhibitor and the acylCoA synthetase inhibitor are intriguing; however, the authors show only one dose despite indicating that multiple doses were tested. Since other CPT1 inhibitors show off target effects the authors should perform a dose response study directly examining CPT1 activity at different doses to confirm that they are only inhibiting beta oxidation. Similarly, the acylCoA synthetase inhibitor induces global changes in lipid metabolism. This may be why both WT and C37KO cells show necrosis compared to cells not treated with drug. It is likely due to nonspecific changes in lipid metabolism rather than any specific effect on exogenous palmitate handling. Again the authors should do a dose response and specifically examine palmitate handling to make this claim.

Point-to-point reply NCOMMS-21-23138-T

'Fatty acid metabolism in aggressive B-cell lymphoma is inhibited by tetraspanin CD37' by Rens Peeters *et al.*

Reviewer #1

In this manuscript, Peeters *et al.* find that CD37, a tetraspanin protein family member associated with poor prognosis in aggressive lymphomas, is a negative regulator of fatty acid (FA) metabolism. Using a combination of lymphoma cell line and mouse knockout models, they show that CD37 deficiency is associated with enhanced palmitate uptake and metabolism, as well as increased lipid droplet (LD) formation. The effect of CD37 on FA metabolism are proposed to be mediated by the inhibitory interaction between CD37 and the FA transporter FATP1 at the plasma membrane. Overall, the studies provide new information on the role of CD37 in lymphomagenesis through regulation of FA metabolism. They also identify FATP1 as a novel molecular effector of CD37 beyond previously identified transporters and signaling pathways. A significant limitation of the current manuscript is the lack of evidence that FATP1 is functionally required for the effect of CD37 deficiency on FA metabolism. In addition, the studies implicating altered LD formation in the context of CD37 deficiency are mostly correlative and underdeveloped.

Specific comments:

1) The current studies lack critical test of FATP1 as the functional link between CD37 and regulation of FA metabolism. The PLA data in support of complex formation between CD37 and FATP1 are intriguing and the downstream manipulation of CPT1 and ACSL1 show FAO is relevant for proliferation and survival of CD37 KO cells but do not directly point to the functional requirement for FATP1 in this setting. The author should test the consequences of FATP1 inhibition/knockdown/knockout on FA metabolism in the context of CD37 negativity. This is important given the many downstream targets of CD37 beyond FATP1 that can regulate FA metabolism. For example, the authors have previously implicated IL-6 signaling in CD37 negative lymphomas. IL-6 can enhance lipolysis and FAO.

We thank the reviewer for these suggestions and performed new metabolic studies with the FATP1 inhibitors arylpiperazine 5k and 12a that have been recently reported as specific inhibitors [Matsufuji, T. *et al. Bioorg. Med. Chem. Lett.* (2013), Mendes, C. *et al. Sci. Rep.* (2019)]. We first validated that these inhibitors did not affect cell viability in our system (new Figure S5A). Our new data demonstrate that FATP1 inhibition was effective in restoring the enhanced FA handling in CD37KO lymphoma cells to wild-type levels in multiple different cell models. Specifically, palmitate-dependent increased spare respiratory capacity in CD37KO cells was abolished by FATP1 inhibition, whereas WT cells were not affected by 5K (new Figure 5A-C).

Figure 5

Palmitate uptake by different CD37KO lymphoma cells (BJAB, OciLy1, OciLy19) was diminished (Figures 5H, S5C-D) in response to FATP1 inhibition, whereas lauric acid uptake remained unaffected (Figure 5i, S5H).

In addition, direct ATP production in response to exogenous palmitate supplementation in BJAB and OciLy8 (CD37-positive) and OciLy19 (CD37-negative) cells was abolished upon FATP1 inhibition, specifically in the CD37-negative cells (new Figure 5G and S5B). In contrast, the CD37-positive lymphoma cells failed to respond to palmitate altogether and as such their energy production was not decreased in response to FATP1 inhibition.

Taken together, FATP1 inhibition restored the metabolic activity and ATP production in response to palmitate in CD37-deficient lymphoma cells, indicating that FATP1 is responsible for the FA metabolic switch in CD37-deficient lymphoma. In line with this, uptake of MCFA (12C lauric acid) that are not transported by FATP1, is not different between CD37KO and WT cells (new Figure 4C, shown under point 3), and MCFA are not depleted from serum of CD37KO mice (Figure 2). Finally, we validated the interaction between CD37 interacts and FATP1 by co-immunoprecipitation experiments (new Figure 4J, K).

2) The studies related to LDs in Fig. 7 are underdeveloped, and do not coherently flow with the rest of the studies in the paper. In their current version, they predominantly provide correlative data. CD37 deficient cells appear to respond to palmitate both via FAO (Fig. 1 and 4) and LD formation (Fig 7b). The idea of LDs as an adaptive mechanism against lipotoxicity is interesting and has been shown in other cell types as well. However, the functional relevance of these observations/ideas in CD37-dependent regulation of substrate switching is unclear and has not been adequately addressed. How relevant is lipotoxicity in the biology of germinal center B cells/lymphomas? Alternatively, are LDs predominantly a reservoir of fuel for these cells? The authors should either provide a more coherent integration of these observations with the rest of the paper or consider removing these data from the current manuscript.

To address these points, we first investigated intracellular lipid droplets (LD) in tissues of patients with DLBCL (n=22). Interestingly, CD37-negative lymphomas not only contained more extracellular large lipid deposits (Figure 7C-F), but also significantly more LD per cell that were bigger in size (new Figure 7G-I).

The lymphoma cells containing these high intracellular LD content were present in close proximity to the large extracellular deposits in the tumour microenvironment of CD37-negative samples (new Figure 7G, arrow). It is possible that these extracellular lipid deposits serve as lipid storage that can be used by CD37-negative lymphoma cells by taking up free fatty acids. In our transcriptome analysis we observed the enzyme phospholipase 2g2d

(*PLA2g2d*) as most significantly upregulated FA-associated gene in CD37-negative cells (**new Figure S7F**). *PLA2g2d* can be secreted to release fatty acids from complex extracellular lipid structures, such as macro-lipid deposits (doi:10.1016/j.celrep.2020.107579). This was confirmed in primary DLBCL samples in published databases, where *PLA2g2D* was found to be highly expressed by DLBCL tissues compared to paired-healthy tissues, whereas this was not the case for any other form of cancer (**new Figure 7SG, H**). Although further studies are needed to unravel the role of this enzyme in the lymphoma microenvironment, this could explain the relation between the macro-lipid deposits and the intracellular LD found in the lymphoma cells, and fits with the recent paper reporting that phospholipases are found to accelerate aggressive B cell lymphoma (Kuda et al., *Cell Metabolism*, 2022). We have added these new data to Figure S7 and deliberate on the implications in the discussion of the revised manuscript.

Thank you for the nice question regarding lipotoxicity and the role of LD. We observed that CD37KO lymphoma cells accumulated LD in time when cultured with excess palmitate (Figure 7B), which was accompanied with decreased viability of WT cells, but not CD37KO cells (Figure S7A,B). In addition, inhibition of LD-associated enzyme fatty acid synthase (FAS) by C75 resulted in more necrosis in CD37KO cultures (Figure S7D). Together, these data indicate that CD37KO lymphoma cells require storage of excess palmitate into LD to prevent cell death possibly caused by lipotoxicity. The observed reduction in ATP availability after 24 hours of FAS inhibition in CD37KO suggests that LDs are also used for long-term fuel storage (Figure S7E). Not much is known about the importance of LDs in germinal centre B cells. An increase in LD numbers is commonly observed in proliferating cells, suggesting it may aid in proliferation and provide fuel storage in time of nutrient deprivation such as during the germinal centre reaction. Clearly, more studies are needed to decipher the role of LD in germinal centre biology, as we elude on in the discussion of the revised manuscript.

3) *The proposed mechanism of CD37's effect on FA metabolism suggests specificity for palmitate within LCFA species. The biochemical basis for this specificity will strengthen the manuscript. Is this specificity strictly dictated by FATP1? This is relevant given the lack of consensus in the field surrounding the exact role of FATP1 in FA metabolism. In addition, the authors should strengthen/validate the selectivity of the proposed mechanism for palmitate by using another FA as a negative control for palmitate in feeding the TCA cycle metabolites (Fig. 4) and FA uptake studies (Fig. 5).*

We investigated CD37 specificity for palmitate and its effects on other lipid species, and found that CD37 does not affect uptake of medium-chain fatty acids (MCFA), such as 12C lauric acid (**new Figure 4C**). This is in line with lipidomic analysis of serum from CD37KO mice, in which LCFA (but not MCFA) are depleted compared to serum from WT mice.

Next, we investigated the contribution of CD37 in processing of oleic acid (18C) by lymphoma cells. Although uptake itself could not be measured (there is no fluorescent analogue available), supplementing CD37KO cells with oleic acid resulted in a small, but non-significant, increase in ATP production compared to WT cells (**new Figure 11**).

In line with this, seahorse analysis revealed a small increase in substrate-induced OCR and spare-respiratory capacity in response to oleic acid, that was not different between WT and CD37KO lymphoma cells (**new Figure S1G-I**).

Taken together, palmitate is clearly the main FA involved in the metabolic switch in CD37KO cells (Figures 1D-I, 2A-E, 3, 5A-F), whereas minor effects were observed for oleic acid, and no effects were found for lauric acid. These data are in agreement with the consensus that FATP1 can transport long-chain fatty acids of different lengths, and with the lipidomic analysis of serum of CD37KO mice showing specific depletion of LCFA, but not of MCFA (Figure 2).

4) The authors selected BJAB and OCyLy1 as human cell line models for CD37 knockout, seahorse and metabolic tracing studies (although only BJAB data are shown/discussed). It will be helpful to include the rationale behind selecting these particular cell lines in terms of their starting CD37 status. As the authors know, certain human DLBCL cell lines readily oxidize palmitate (Pfeiffer, K422, etc). Are these lines negative for CD37? Is gain-of-function in CD37 sufficient to block FAO in these DLBCL cell lines?

BJAB and OCiLy1 were selected as human cell models, since CD37KO clones were generated previously for these cell lines (BJAB: Elfrink et al. *Blood* 2019). We agree that the study would be strengthened by inclusion of different DLBCL cell lines. Although we did not have availability of the above mentioned cell lines, we extended our studies to multiple DLBCL cell lines that have been recently reported to be endogenously CD37-positive or CD37-negative (Elfrink et al. *Blood Adv* 2022). We observed different CD37-positive DLBCL cell lines (BJAB, OciLy8, DOHH2, WSU-NHL) to be non-responsive to palmitate in Seahorse analysis and ATP production, in contrast to CD37-negative DLBCL cell lines (BJAB-CD37KO, SUDHL6, OciLy19) that showed increased SRC and ATP production after palmitate supplementation (**new Figure S4A-L**). Moreover, these effects were largely blocked by 2 different FA metabolism inhibitors (ST1326: new Figure S6G-N, etomoxir).

Thus, these data show a strong correlation between CD37 status and the fatty acid metabolism response, indicating that CD37 function in blocking FA metabolism is conserved among human DLBCL cell lines.

5) Related to the above point, what is the FAO capacity of the CD37 negative primary DLBCLs in Figure 7 compared to CD37 positive counterparts?

Since we have only availability of tissue samples of primary DLBCL, our studies on patient material are limited to immunohistochemistry. We provide new data on the correlation between CD37 status of the primary DLBCL and the amount/sized of lipid droplets (new Figure 7G-I, see above). LDs are associated with chemoresistance and with cancer progression. Interestingly, the paper which reported CD37 to be an independent prognostic factor for DLBCL in a large cohort (n=806), also included transcriptome analysis (Xu-Monette et al, *Blood* 2016). *ACSL1* was among the most upregulated genes within CD37-negative DLBCL compared to CD37-positive DLBCL. We observed ACSL1 inhibition to be particularly effective in CD37KO lymphoma cells (Figure 6G). Furthermore, gene expression profile analysis of different primary lymphomas revealed that overexpression of FATP1 as most significant predictor for aggressiveness (Magi et al., *Sci Rep.*, 2019). Thus, although limited, these studies are in line with enhanced FAO capacity in aggressive, CD37-negative DLBCL. We discuss these findings in the discussion of the revised paper.

Additional points:

6) Fig. 6 panels g, h and j- CPT1 and ACSL1 inhibition also have substantial effect on viability, proliferation and necrosis of WT cells. The authors should include stats for these comparisons as well. All stats are now included and we also provided full concentration gradients for both inhibitors (new Figure 6).

7) A rationale for why the authors specifically measured necrotic death (as opposed to other forms of cell death) will improve the flow of the text. It is also unclear why the authors chose different readouts for CPT1 vs ACSL1 inhibition (survival and proliferation for one, ATP and necrosis for the other). The description of these results

(lines 250-251) is also confusing as the effect of CPT1 and ACSL1 inhibition on necrosis is compared in the text but Fig. 6j only involves ACSL1 inhibition.

We have included all readouts for both inhibitors (survival, ATP production: new Figure 6) and apoptosis/necrosis is presented in new Figure S6A-F.

8) It appears that in all Seahorse experiments, the media contains 2 mM glutamine at minimum. As such, referring to this as nutrient deprived condition does not seem accurate.

We fully agree and have changed this to 'nutrient-restricted', as it is markedly more restricted compared to full growth medium in which the cells are cultured.

Reviewer #2

In this study, the authors provide evidence that CD37-negative B cell lymphomas undergo a metabolic switch from glycolysis to fatty acid oxidation, in particular by using palmitate as an energy source. Supporting their hypothesis, they provide extensive metabolomic data on CD37-negative and control B cell lymphomas, and data supporting an interaction between CD37 and the fatty acid cell surface transporter, FATP1. The potential impact of the study is high because it would lay the foundation for understanding the aggressive behavior of CD37-negative B cell lymphomas and point to potential therapeutic interventions targeting fatty acid utilization as an energy source. There are a few major issues that need to be addressed to make the case for the central working hypothesis advanced by the paper.

1. Evidence for a direct CD37-FATP1 interaction is based only on proximity ligation assay (PLA). It is unclear that PLA has the spatial resolution to distinguish direct from indirect protein-protein interactions, but the authors repeatedly claim it as evidence for a direct interaction.

PLA provides information on protein-protein proximity within 40 nanometre scale at the cell surface. Although this is expected to reveal direct protein interactions, we agree this technique does not fully prove direct interaction, and therefore we rephrased the text into 'indicative of interaction'. In addition, we confirmed the interaction between CD37 and FATP1 by co-immunoprecipitation (discussed below).

2. The PLA assay needs additional controls. While the fact that no PLA signal is seen between CD37 and FATP1 antibodies in CD37-negative cells is reassuring regarding non-specific binding of the CD37 antibody, a stronger negative control is required. For example, showing that a GPI-linked protein such as CD55 does not generate a PLA signal with CD37 or FATP1, would increase confidence in the specificity of the assay. Alternatively, demonstrating the CD37-FATP1 interaction by a second additional method would increase confidence in the interaction.

We thank the reviewer for this suggestion and confirmed the CD37-FATP1 interaction by an alternative method. FATP1 was co-immunoprecipitated with CD37 upon transfecting FATP1-FLAG and CD37-alpha-TAG in CD37KO lymphoma cells (**new Figure 4J**: FATP1 band at ~63 Kd). Controls (mock transfected cells, or cells transfected with only CD37 or FATP1), did not result in a FATP1 signal when stained against FLAG-tag. Three independent co-IP experiments were performed verifying the CD37-FATP1 interaction. This, together with the PLA experiments visualizing endogenous CD37 and FATP1 on the lymphoma cell surface, confirm that CD37 and FATP1 interact.

3. The authors claim that CD37 negatively regulates FATP1 activity. This is an appealing hypothesis, but no direct evidence is provided to support it. Uptake of palmitate analog Bodipy FL C16 is reduced in CD37-positive cells, but the proportion of the uptake mediated by FATP1 was not measured. Moreover, to make a strong claim that

CD37 is functioning by directly regulating FATP1, something like expressing a FATP1 mutant incapable of CD37 interaction or some other type of structure-function experiment would be required.

To substantiate the finding that FATP1 is the functional link between CD37 and FA metabolism, we performed new metabolic studies with the FATP1 inhibitors 5K and 12K that have been recently reported as specific inhibitors [Matsufuji, T. *et al. Bioorg. Med. Chem. Lett.* (2013), Mendes, C. *et al. Rep.* (2019)]. We first validated that these inhibitors did not affect cell viability in our system (**new Figure S5A**). Our new data demonstrate that FATP1 inhibition was effective in restoring the enhanced FA handling in CD37KO lymphoma cells to wild-type levels in multiple different cell models. Specifically, palmitate-dependent increased spare respiratory capacity in CD37KO cells was abolished by FATP1 inhibition, whereas WT cells were not affected by 5K (**new Figure 5A-C**).

Figure 5

Palmitate uptake by different CD37KO lymphoma cells (BJAB, OciLy1, OciLy19) was diminished (new Figures 5H, S5C-D) in response to FATP1 inhibition, whereas lauric acid uptake remained unaffected (new Figure 5i, S5H).

In addition, direct ATP production in response to exogenous palmitate supplementation in BJAB and OciLy8 (CD37-positive) and OciLy19 (CD37-negative) cells was abolished upon FATP1 inhibition, specifically in the CD37-negative cells (**new Figure 5G, S5B**). In contrast, the CD37-positive lymphoma cells failed to respond to palmitate altogether and as such their energy production was not decreased in response to FATP1 inhibition.

Taken together, FATP1 inhibition restored the metabolic activity and ATP production in response to palmitate in CD37-deficient lymphoma cells, indicating that FATP1 is responsible for the FA metabolic switch in CD37-deficient lymphoma. In line with this, uptake of medium chain fatty acids (MCFA, such as 12C lauric acid) that are not transported by FATP1, is not different between CD37KO and WT cells (Figure 4C), and MCFA are not depleted from serum of CD37KO mice (Figure 2).

Some additional issues include

4. In some statistical analyses, WT control data are flattened to equal exactly 1.0, but then used for statistical testing (Fig. 3f,g,h, Fig. 5a and Fig. 7a). The t-test is influenced by the spread in the data, but this has been artificially removed for the WT control data in these panels. The WT control data points could each be divided by

WT control overall mean to restore the variability in the WT control data points to what was actually observed in the experiments, and then the t-test could be used.

We fully agree, we corrected this for all the graphs in the revised manuscript. For the carnitine tracing (Figure 2G-I), this was not possible since every individual point represents a single acyl-carnitine, with different groups and modifications. Because HPLC only provides relative abundance per individual metabolite, differences can only be expressed as relative increase or decrease within the same carnitine species between different treated groups.

5. In Fig. 5F, how is it possible to have a negative number of counts per second in luminescence assay for ATP? This was relative ATP accumulation compared to non-treated (NT), so if the non-treated samples produced more ATP than the treated samples, the relative ATP was negative. To prevent confusion, we changed the label of the y-axis to 'ATP (CPS relative to NT)' (Figure 2E, 5G).

6. In the IHC data in Fig. 7c&d, it's not clear how the large fatty deposits in the CD37-negative tissue specimen (many times larger than individual cells?) relate to lipid droplets that accumulate inside cells.

To address this question, we investigated intracellular lipid droplets (LD) in tissues of patients with DLBCL (n=22). Interestingly, CD37-negative lymphomas not only contained more extracellular large lipid deposits (Figure 7C-F), but also significantly more LD per cell that were bigger in size (new Figure 7G-I).

The lymphoma cells containing these high intracellular LD content were present in close proximity to the large extracellular deposits in the tumour microenvironment of CD37-negative samples (Figure 7G, arrow). It is possible that these extracellular lipid deposits serve as lipid storage that can be exploited by CD37-negative lymphoma cells by taking up free fatty acids. In our transcriptome analysis we observed the enzyme phospholipase 2g2d (*PLA2g2d*) as most significantly upregulated gene in CD37-negative cells (new Figure S7F). *PLA2g2d* can be secreted to release fatty acids from complex extracellular lipid structures, such as macro-lipid deposits (doi:10.1016/j.celrep.2020.107579). This was confirmed in primary DLBCL samples in published databases, where *PLA2g2D* was found to be highly expressed by DLBCL tissues compared to paired-healthy tissues, whereas this was not the case for any other form of cancer (new Figure 7SG, H). Although further studies are needed to unravel the role of this enzyme in the lymphoma microenvironment, this could explain the relation between the macro-lipid deposits and the intracellular LD found in the lymphoma cells, and fits with the recent paper reporting that phospholipases are found to accelerate aggressive B cell lymphoma (Kuda et al., *Cell Metabolism*, 2022). We have added these new data to Figure S7 and deliberate on the implications in the discussion of the revised manuscript.

Reviewer #3

This manuscript by Peeters et al. examines metabolic alterations in nutrient utilization in B cell lymphoma induced by loss of the tetraspanin, CD37. This group has previously shown that loss of CD37 expression in this cancer leads to a poorer prognosis and this makes the clinical relevance of this manuscript high. However, some of the major metabolic findings of the manuscript show high variability and refinement of data presentation is needed.

Major comments:

1. The authors perform lipidomics (Fig 2) on CD37 KO mice and WT mice with and without B cell lymphoma. Examining the serum, they don't see overall changes in serum lipid levels but examination of individual species shows changes in TAGS and FFA. However, when FFA were quantified only palmitate shows a statistically significant difference. From the heat maps shown in figure 2 there appears to be significant variability between biological replicates especially WT CD37KO without tumor. The lack of correlation between the lipid quantification and the fold changes shown in Figure 2d are confusing.

We agree that the use of fold change in heatmaps to display a dataset with a wide range of absolute concentrations results in the loss of information. Therefore, we removed the heatmaps and now present absolute concentrations (**new Figure 2A-C**). Moreover, we repeated the lipidomic analysis in new experiments to increase the number of biological replicates of healthy serum from n=3 to n=6 per genotype. The new data confirm the decreased total FFA levels in serum of CD37KO mice compared to WT mice. Palmitate is not the only LCFA that is depleted from the serum of healthy CD37KO mice, oleic acid and linoleic acid are also depleted. However, palmitate is most severely depleted, and the only LCFA that is depleted in a tumour-dependent manner within the CD37KO mice:

All raw lipidomics data can be accessed in the raw data file or at metabolights: <https://www.ebi.ac.uk/metabolights/MTBLS4978>.

Login: rens.peeters@radboudumc.nl, PW: 475a01

FFA and lipids in serum are heavily dependent on diet and no methods are described for controlling whether the mice were fed or fasted prior to the lipidomics analysis and whether the large tumor burden in the CD37KO impacted their ability to eat (do they exhibit weight loss over the course of disease). It seems unlikely that there would be preferential uptake of palmitate for beta - oxidation versus other long chain fatty acids as they are equally likely to enter into this pathway. There are also significant differences in other LCFA just examining CD37KO animals to WT. Could the authors resolve this discrepancy with additional biological replicates and control of feeding?

The WT (CD37+/+ littermate controls) and CD37KO mice were housed in the same room and both strains had free access to *ad libitum* chow and water. CD37KO mice did not show a different eating pattern and both strains

have normal growth and weight (**new Figure S2**). Moreover, several lipid species remain unaffected or were even increased in serum of CD37KO tumour-bearing mice (Figure 2A), making it unlikely that the observed difference in LCFA is caused by an underlying different eating pattern.

2. Similarly, in the isolated B cells there is the same mismatch between the heat maps and lipid quantification. This again may be due to variability between biological replicates, particularly since one CD37KO sample seems to show larger increases in C16, C20, C20:4, and C18 compared to other replicates and this may skew the results. Since the authors show that CD37 interacts with FATP1 which is a general LCFA transporter it seems unlikely that there is specificity towards palmitate. The overall changes in total FFA shown in 3C likely reflect changes in the levels of other lipids that aren't captured in the statistical analysis of quantification of individual species due to variability.

We agree and have resolved the issue by replacing heatmaps with graphs showing absolute compositions. Moreover, we repeated the lipidomic analysis in new experiments to double the number of biological replicates from n=3 to n=6 (**new Figure S2B-C**). In order to avoid normalization issues between the two individual runs, we reported everything in composition (%) instead of absolute concentrations. The pooled data show no differences in FFA between healthy B cells of WT and CD37KO mice. The abundance of the different FAs at a single given time is the result of a balance between uptake/synthesis on the one hand and breakdown/storage on the other hand. Therefore, we also assessed whether these FA uptake and breakdown were different between primary WT and CD37KO B cells. CD37KO B cells have both enhanced palmitate uptake and breakdown potential compared to WT B cells (**Figure 2D-F**).

3. The authors claim that the changes in short chain and medium chain acylcarnitines reflects increased beta oxidation may not be correct. Additionally, the schematic in 4B is incorrect. Short chain and medium fatty acids do not require conjugation to carnitine to be transported into the mitochondria, rather they diffuse through the membrane. Carnitine conjugation is used for transport into the mitochondria and once in the mitochondria acylCoA species are formed via CPT2. The acylCoA species then undergo beta oxidation. It is incorrect to show that the acylcarnitine species undergo beta oxidation in figure 4B. Are these excess SCFA and MCFA carnitine species just a byproduct of excess lipids in CD37KO cells? Fact that label is present has to come from palmitate.

We adjusted the flux schematic to show proper carnitine conjugation prior to mitochondrial entrance (new Fig. 3A). We now also show total levels and the full isotopologue pattern of the SCFA butyrate to support our hypothesis that this SCFA is at least in part produced via FAO. As pointed out by the reviewer also, the heavy carbons in the butyric acid per definition must have originated from ^{13}C -palmitate. We observed that the sum of all labelled isotopologues (M+2 and higher) butyryl-L-carnitine is significantly higher in CD37KO cells compared to WT cells after 4 hours, even though total levels of butyryl-L-carnitine are similar between WT and CD37KO (new Figure 3F-I). This means that a larger fraction of butyrate is formed from labelled palmitate via FAO in CD37KO cells compared to WT, making it unlikely that excess SCFA carnitine species are reflection of excess lipids in CD37KO cells only.

4. The isotope tracing experiments shown in Figure 4 and Supplemental Figure 3 are confusing and the data presentation is not clear. The authors should show full isotopologue patterns for TCA metabolites and palmitoylcarnitine as these are the primary metabolites that should be labeled when cells are fed labeled carnitine. Additionally, showing fold changes in heat maps instead of labeling patterns is confusing and not typically how isotope labeling experiments are presented. Are the authors suggesting that labeled butyryl carnitine arises from incomplete oxidation of palmitate in the mitochondria and subsequent conjugation to carnitine? The transport of acylcarnitines into mitochondria is driven by favorable free energy changes due to beta oxidation; ensuring transport of FA into mitochondria and complete oxidation. If the authors are proposing that labeling of SCFA-carnitines are occurring due to incomplete oxidation they need to show that.

We agree and now show full labelling patterns for the metabolites of interest. We do still present the significance of the sum of M+2 isotope labelled metabolites, since this is a good indicator to determine how much of the metabolite is synthesized as a direct result from acetyl-CoA from palmitate feeding into the TCA and gives insights in what benefits the enhanced FA metabolism can provide for CD37KO cells. The heatmaps show the fold change in the sum of the peak area of all non-labeled and labeled isotopologues combined (M+0 and higher), providing a good overview of what metabolic pathways are prioritized by the different genotypes. We do believe that the labelled SCFA are indeed a consequence of FAO of exogenous palmitate, because this is the only way the SCFA could obtain heavy carbons, as discussed above. Others have previously shown that SCFAs may accumulate when active FAO is engaged in the presence of non-limiting amounts of palmitate, as the same enzyme is responsible for the first hydrolysis steps. Since palmitate is continuously transported into the mitochondria and degraded, the relative amount of partially oxidized acyl moieties increases. Because SCFA can passively diffuse out of the mitochondria, just by chance there will be 'leak' of these SCFAs that is indicative of active FAO. Others have reported increased accumulation of SCFAs in cells with overexpressed CPT1a and FATP1 (Sebastián et al., *J Lipid Res.*, 2009).

5. The functional analyses with the CPT1a inhibitor and the acylCoA synthetase inhibitor are intriguing; however, the authors show only one dose despite indicating that multiple doses were tested. Since other CPT1 inhibitors show off target effects the authors should perform a dose response study directly examining CPT1 activity at different doses to confirm that they are only inhibiting beta oxidation. Similarly, the acylCoA synthetase inhibitor induces global changes in lipid metabolism. This may be why both WT and C37KO cells show necrosis compared to cells not treated with drug. It is likely due to nonspecific changes in lipid metabolism rather than any specific

effect on exogenous palmitate handling. Again the authors should do a dose response and specifically examine palmitate handling to make this claim.

We included full concentration gradients for the CPT1a and ACSL1 inhibitors and their effect on viability, apoptosis, necrosis, proliferation and energy production (**new Figure 6, S6A-C**) together with an alternative, more established, inhibitor (etomoxir) in our studies, and provide new metabolic data on multiple CD37-positive and CD37-negative lymphoma cell lines (new Figures S1, S4, S6). Finally, we also show a dose-dependent inhibition of palmitate handling that is significantly more affected in the CD37KO cells:

REVIEWERS' COMMENTS

Reviewer #1 (Remarks to the Author):

The authors have addressed my previous comments and the revised manuscript has been strengthened.

Reviewer #2 (Remarks to the Author):

In this revised manuscript, the authors solidify the evidence that CD37-negative B cell lymphomas undergo a metabolic switch from glycolysis to fatty acid oxidation, by using palmitate as an energy source. They now provide strong evidence that the molecular mechanism of this gatekeeper role of CD37 is via molecular interaction with FATP1, a fatty acid cell surface transporter that mediates uptake of palmitate. Bolstering proximity ligation assay results indicating CD37 forms a complex with FATP1, they now provide evidence via co-immunoprecipitation that CD37 interacts with FATP1. These co-IPs were performed in Brij97 detergent, which is a relatively stringent condition for tetraspanin interactions, and FATP1 was only brought down when CD37 was expressed. Importantly, the authors now provide functional data showing that several metabolic effects produced by CD37 deletion are specifically reversed upon pharmacological inhibition of FATP1, providing evidence that FATP1 is the major mediator of the altered metabolism in CD37-deficient cells. The impact of the study is high because it lays the foundation for understanding the aggressive behavior of CD37-negative B cell lymphomas and points to potential therapeutic interventions targeting fatty acid utilization as an energy source. Several minor issues have also been addressed, as well as clarification of the potential significance of large extracellular fatty deposits in CD37-negative tissue specimens.

Reviewer #3 (Remarks to the Author):

The authors have been responsive to comments and the new manuscript is much improved with respect to the lipidomics and the drug studies of FAO inhibition. The only comment that remains is concerning the data display for the serum lipidomics (Figure 2). Showing the absolute concentrations improves data clarity, however, it is unusual that serum concentrations are displayed as nmol/g because the metabolite extraction was from a volume of serum not a mass of tissue. The authors should clarify how these concentrations were determined in the methods.

Subject

Response to referee #3 NCOMMS-21-23138B

Nijmegen, July 14, 2022

Reviewer #3:

The authors have been responsive to comments and the new manuscript is much improved with respect to the lipidomics and the drug studies of FAO inhibition. The only comment that remains is concerning the data display for the serum lipidomics (Figure 2). Showing the absolute concentrations improves data clarity, however, it is unusual that serum concentrations are displayed as nmol/g because the metabolite extraction was from a volume of serum not a mass of tissue. The authors should clarify how these concentrations were determined in the methods.

Answer: The original Lipidyzer platform based its calculations on mass (g), hence nmol/g. A detailed description of the approach can be found here: *Anal Chem.* 2021: 93(49):16369. Nevertheless, as plasma has a density of 1.025g/mL these data are in practice 1:1 translated into nmol/mL. A detailed description of all calculations can be found here: *J Am Soc Mass Spectrom.* 2021: 32(11):2655. We cited these references and clarified this in the methods section.

We thank the reviewers for their constructive feedback that has significantly improved the manuscript, and hope that the manuscript is now acceptable for publication in *Nature Communications*.

Yours sincerely,

Annemiek B. van Sriel, PhD
Professor of Experimental Immunology